# Scalable Multi-Agent Reinforcement Learning for Networked Systems with Average Reward

**Guannan Qu**
Caltech, Pasadena, CA
gqu@caltech.edu

**Yiheng Lin**
Tsinghua University, Beijing, China
linyh16@mails.tsinghua.edu.cn

**Adam Wierman**
Caltech, Pasadena, CA
adamw@caltech.edu

**Na Li**
Harvard University, Cambridge, MA 02138
nali@seas.harvard.edu

## Abstract

It has long been recognized that multi-agent reinforcement learning (MARL) faces significant scalability issues due to the fact that the size of the state and action spaces are exponentially large in the number of agents. In this paper, we identify a rich class of networked MARL problems where the model exhibits a local dependence structure that allows it to be solved in a scalable manner. Specifically, we propose a Scalable Actor-Critic (SAC) method that can learn a near optimal localized policy for optimizing the average reward with complexity scaling with the state-action space size of local neighborhoods, as opposed to the entire network. Our result centers around identifying and exploiting an exponential decay property that ensures the effect of agents on each other decays exponentially fast in their graph distance.

## 1 Introduction

As a result of its impressive performance in a wide array of domains such as game play (Silver et al., 2016; Mnih et al., 2015), robotics (Duan et al., 2016), and autonomous driving (Li et al., 2019), Reinforcement Learning (RL) has emerged as a promising tool for decision and control and there has been renewed interest in the use of RL in multi-agent systems, i.e., Multi-Agent RL (MARL).

The multi-agent aspect of MARL creates additional challenges compared with single agent RL. One core challenge is scalability. Even if individual agents' state or action spaces are small, the global state space or action space can take values from a set of size exponentially large in the number of agents. This "curse of dimensionality" renders the problem intractable in many cases. For example, RL algorithms such as temporal difference (TD) learning or $Q$-learning require storage of a $Q$-function (Bertsekas and Tsitsiklis, 1996) whose size is the same as the state-action space, which in MARL is exponentially large in $n$. Such scalability issues have been observed in the literature in a variety of settings, including Blondel and Tsitsiklis (2000); Papadimitriou and Tsitsiklis (1999); Zhang et al. (2019); Kearns and Koller (1999); Guestrin et al. (2003).

To address the issue of scalability, a promising approach that has emerged in recent years is to exploit problem structure, e.g., (Gu et al., 2020; Qu and Li, 2019; Qu et al., 2019). One promising form of structure is enforcing local interactions, i.e., agents are associated with a graph and they interact only with nearby agents in the graph. Such local interactions are common in networked systems, including epidemics (Mei et al., 2017), social networks (Chakrabarti et al., 2008; Llas et al., 2003), communication networks (Zocca, 2019; Vogels et al., 2003), queueing networks (Papadimitriou and Tsitsiklis, 1999), smart transportation (Zhang and Pavone, 2016), smart building systems (Wu et al.,

2016; Zhang et al., 2017). One powerful property associated with local interactions is the so-called *exponential decay property* (Qu et al., 2019), also known as the *correlation decay* (Gamarnik, 2013; Gamarnik et al., 2014) or *spatial decay* (Bamieh et al., 2002; Motee and Jadbabaie, 2008), which says that the impact of agents on each other decays exponentially in their graph distance. The exponential decay property often leads to potential for scalable, distributed algorithms for optimization and control (Gamarnik, 2013; Bamieh et al., 2002; Motee and Jadbabaie, 2008), and has proven effective for MARL, e.g., Qu et al. (2019).

While exploiting local interactions and exponential decay in MARL has proven effective, results so far have been derived only in cases considering discounted total reward as the objective, e.g., (Qu et al., 2019). This is natural since results focusing on average reward, i.e., the reward in stationarity, are known to be more challenging to derive and require different techniques, even in the single agent RL setting (Tsitsiklis and Van Roy, 1999). However, in many networked system applications, the average reward is a more natural objective. For example, in communication networks, the most common objective is the performance of the system (e.g. throughput) in stationarity.

In this paper, our goal is to derive results for average reward MARL in networked systems. However, it is unclear whether it is possible to obtain results that parallel those of Qu et al. (2019), which focuses on discounted reward. In the average reward case, the exponential decay property exploited by Qu et al. (2019) will no longer be true in general because the average reward case can capture certain NP-hard problems in the worst case. For example, Blondel and Tsitsiklis (2000); Whittle (1988); Papadimitriou and Tsitsiklis (1999) all point out that such Markov Decision Process (MDP) with product state and action spaces is combinatorial in nature and is intractable from a computational complexity perspective in the worst case.

**Contributions.** Despite the worst case intractability results, in this paper we show there are large classes of networked systems where average reward MARL is tractable. More specifically, our main technical result (Theorem 1) shows that an exponential decay property still holds in the average reward setting under certain conditions that bound the interaction strength between the agents. These conditions are general and, numerically, exponential decay holds with high probability when the problem instance is generated randomly. Given the presence of exponential decay, we develop a two time-scale actor-critic method (Konda and Tsitsiklis, 2000) and show that it finds a localized policy that is a $O(\rho^{\kappa+1})$-approximation of a stationary point of the objective function (Theorem 2) with complexity that scales with the local state-action space size of the largest $\kappa$-hop neighborhood, as opposed to the global state-action space size. To the best of our knowledge, this is the first such provable guarantee for scalable MARL with average reward for networked systems. Finally, to demonstrate the effectiveness of our approach we illustrate its performance in a setting motivated by protocol design for multi-access wireless communication.

The key analytic technique underlying our results is a novel MDP perturbation result. We show that, when the local interaction between nodes is bounded, a perturbation on the state of an agent has diminishing affect on the state distribution of far away agents, which enables us to establish the exponential decay property that enables tractable learning. Our result has a similar flavor as some static problems in theoretic computer science, like the Lovász local lemma (Moser and Tardos, 2010); or the "correlation decay" in Gamarnik et al. (2014) for solving combinatorial optimization problems. However, our result handles the more challenging dynamic setting where the states evolve over time.

**Related Literature.** MARL dates back to the early work of Littman (1994); Claus and Boutilier (1998); Littman (2001); Hu and Wellman (2003) (see Bu et al. (2008) for a review) and has received considerable attention in recent years, e.g. Zhang et al. (2018); Kar et al. (2013); Macua et al. (2015); Mathkar and Borkar (2017); Wai et al. (2018) and the review in Zhang et al. (2019). MARL considers widely-varying settings, including competitive agents and Markov games; however the setting most relevant to ours is cooperative MARL. In cooperative MARL, the typical setting is that agents can choose their own actions but they share a common global state (Bu et al., 2008). In contrast, we study a more structured setting where each agent has its own state that it acts upon. Despite the different setting, cooperative MARL problems still face scalability issues since the joint-action space is exponentially large. Methods have been proposed to deal with this, including independent learners (Claus and Boutilier, 1998; Matignon et al., 2012), where each agent employs a single-agent RL method. While successful in some cases, such method can suffer from instability (Matignon et al., 2012). Alternatively, one can approximate the large $Q$-table through linear function approximation (Zhang et al., 2018) or neural networks (Lowe et al., 2017). Such methods can reduce computation

complexity significantly, but it is unclear whether the performance loss caused by the function approximation is small. In contrast, our technique not only reduces computation but also guarantees small performance loss.

Our work adds to the growing literature that exploits exponential decay in the context of networked systems. The exponential decay concept we use is related to the concept of "correlation decay" studied in Gamarnik (2013); Gamarnik et al. (2014), though their focus is on solving static combinatorial optimization problems whereas ours is on learning polices in dynamic environments. Most related to the current paper is Qu et al. (2019), which considers the same networked MARL setting but focuses on discounted reward. The discount factor ensures that exponential decay always holds, in stark contrast to the more challenging average setting which is combinatorial in nature and is intractable in the worst case. Additionally, our result in the average reward case improves the bound on the exponential decay rate in the discounted setting (Corollary 1).

More broadly, this paper falls under the category of "succinctly described" MDPs in Blondel and Tsitsiklis (2000, Section 5.2), which shows that when the state/action space is a product space formed by the individual state/action spaces of multiple agents, the resulting Markov Decision Process (MDP) is intractable in general, even when the problem has structure (Blondel and Tsitsiklis, 2000; Whittle, 1988; Papadimitriou and Tsitsiklis, 1999). In the context of these worst case complexity results, our work identifies conditions when it is possible to develop scalable methods.

Finally, our work is related to *factored MDPs*, *weakly coupled MDPs*, and *mean-field RL*, though the model and the focus are very different. In factored MDPs, there is a global action affecting every agent whereas in our case, each agent has its own action (Kearns and Koller, 1999; Guestrin et al., 2003; Osband and Van Roy, 2014). In weakly coupled MDPs, agents' transitions are decoupled (Meuleau et al., 1998) in stark contrast to our coupled setting. In mean-field RL, the agents are coupled but are assumed to be "homohenous", therefore the influence among agents can be well approximated by their average (Yang et al., 2018; Gu et al., 2020). In contrast, our work does not place homogeneity assumptions, but assumes local interactions instead.

## 2  Preliminaries

In this section, we formally introduce the problem and define a key concept, the exponential decay property, that will be used throughout the rest of the paper.

### 2.1  Problem Formulation

We consider a network of $n$ agents that are associated with an underlying undirected graph $\mathcal{G} = (\mathcal{N}, \mathcal{E})$, where $\mathcal{N} = \{1, \ldots, n\}$ is the set of agents and $\mathcal{E} \subset \mathcal{N} \times \mathcal{N}$ is the set of edges. Each agent $i$ is associated with state $s_i \in \mathcal{S}_i$, $a_i \in \mathcal{A}_i$ where $\mathcal{S}_i$ and $\mathcal{A}_i$ are finite sets. The global state is denoted as $s = (s_1, \ldots, s_n) \in \mathcal{S} := \mathcal{S}_1 \times \cdots \times \mathcal{S}_n$ and similarly the global action $a = (a_1, \ldots, a_n) \in \mathcal{A} := \mathcal{A}_1 \times \cdots \times \mathcal{A}_n$. At time $t$, given current state $s(t)$ and action $a(t)$, the next individual state $s_i(t+1)$ is independently generated and is only dependent on neighbors:

$$P(s(t+1)|s(t), a(t)) = \prod_{i=1}^{n} P_i(s_i(t+1)|s_{N_i}(t), a_i(t)), \tag{1}$$

where notation $N_i$ means the neighborhood of $i$ (including $i$ itself) and $s_{N_i}$ is the states of $i$'s neighbors. In addition, for integer $\kappa \geq 1$, we let $N_i^{\kappa}$ denote the $\kappa$-hop neighborhood of $i$, i.e. the nodes whose graph distance to $i$ is less than or equal to $\kappa$, including $i$ itself. We also use $z = (s, a)$ and $\mathcal{Z} = \mathcal{S} \times \mathcal{A}$ to denote the state-action pair (space), and $z_i$, $\mathcal{Z}_i$ are defined analogously.

Each agent is associated with a class of localized policies $\zeta_i^{\theta_i}$ parameterized by $\theta_i$. The localized policy $\zeta_i^{\theta_i}(a_i|s_i)$ is a distribution on the local action $a_i$ conditioned on the local state $s_i$, and each agent, conditioned on observing $s_i(t)$, takes an action $a_i(t)$ independently drawn from $\zeta_i^{\theta_i}(\cdot|s_i(t))$. We use $\theta = (\theta_1, \ldots, \theta_n)$ to denote the tuple of the localized policies $\zeta_i^{\theta_i}$, and use $\zeta^{\theta}(a|s) = \prod_{i=1}^{n} \zeta_i^{\theta_i}(a_i|s_i)$ to denote the joint policy.

Each agent is also associated with a stage reward function $r_i(s_i, a_i)$ that depends on the local state and action, and the global stage reward is $r(s, a) = \frac{1}{n} \sum_{i=1}^{n} r_i(s_i, a_i)$. The objective is to find

localized policy tuple $\theta$ such that the global stage reward in stationarity is maximized,

$$\max_\theta J(\theta) := \mathbb{E}_{(s,a)\sim\pi^\theta} r(s,a), \tag{2}$$

where $\pi^\theta$ is the distribution of the state-action pair in stationarity. We also define $J_i(\theta)$ to be $J_i(\theta) = \mathbb{E}_{(s,a)\sim\pi^\theta} r_i(s_i, a_i)$, which satisfies $J(\theta) = \frac{1}{n}\sum_{i=1}^n J_i(\theta)$.

To provide context for the rest of the paper, we now review a few key concepts in RL. Fixing a localized policy $\theta$, the $Q$ function for the policy is defined as,

$$
\begin{aligned}
Q^\theta(s,a) &= \mathbb{E}_\theta\left[\sum_{t=0}^\infty \big(r(s(t),a(t)) - J(\theta)\big)\Big| s(0)=s, a(0)=a\right]\\
&= \frac{1}{n}\sum_{i=1}^n \mathbb{E}_\theta\left[\sum_{t=0}^\infty \Big(r_i(s_i(t),a_i(t)) - J_i(\theta)\Big)\Big| s(0)=s, a(0)=a\right] := \frac{1}{n}\sum_{i=1}^n Q_i^\theta(s,a), \quad (3)
\end{aligned}
$$

where in the last step, we have also defined $Q_i^\theta$, the $Q$-function for the local reward. We note here that both $Q^\theta$ and $Q_i^\theta$ depends on the state-action pair of the whole network, and is thus intractable to compute and store.

Finally, we review the policy gradient theorem (Lemma 1), an important tool for our analysis. It can be seen that the exact policy gradient relies on the full $Q$-function, which may be intractable to compute due to its large size.

**Lemma 1** (Sutton et al. (2000)). *Recall $\pi^\theta$ is the stationary distribution for the state-action pair under policy $\theta$. The gradient of $J(\theta)$ is then given by*

$$\nabla_\theta J(\theta) = \mathbb{E}_{(s,a)\sim\pi^\theta} Q^\theta(s,a)\nabla_\theta \log \zeta^\theta(a|s).$$

*Average reward vs. discounted reward.* Our objective (2) is the average reward under the stationary distribution, which is different from the more commonly studied $\gamma$-discounted reward, $\mathbb{E}\sum_{t=0}^\infty \gamma^t r(s_t, a_t)$ for some $\gamma \in (0,1)$, starting from some fixed initial state $s_0$. In many applications, there is no naturally defined starting state or discounting factor, and the performance is measured under the stationary distribution. For example, in wireless communication (Section 4), the long standing performance metric is the throughput, which measures the average number of packets sent under the stationary distribution. Despite its relevance in applications, it is widely known that average reward RL is more challenging than discounted reward RL even in single-agent settings. For example, in average reward RL, the Bellman operator is no longer a contraction in general, and the set of fixed points is no longer unique compared to the discounted case (Tsitsiklis and Van Roy, 1999, 2002). On top of the above, in multi-agent settings, the average reward case faces additional challenges. Specifically, as will be shown in Appendix A.2, the average reward problem captures certain NP-hard instances, in sharp contrast to the discounted case in Qu et al. (2019) where the hardness result does not hold.

*Comparison to mean-field RL.* A highly related problem to our setting is mean-field multi-agent RL (Gu et al., 2020), where similar to our setting, the global state and action space is the product space of local state and action spaces, leading to the same curse of dimensionality issue as our problem. However, different to our local interaction structure (1), Gu et al. (2020) allows global interaction but assumes the agents are identical, and only interact through "population distribution", meaning each agent's transition depends on other agents only through other agents' aggregate state and action distribution. This difference in structural assumption makes the application scenario, algorithm design and analysis of Gu et al. (2020) very different from ours, and we leave it as future work to compare Gu et al. (2020) with our work in more detail.

## 2.2 Exponential Decay Property and Efficient Approximation of $Q$-Function

Given the local interaction structure in the probability transition (1), a natural question to ask is that whether and under what circumstance, nodes that are far away have diminishing effect on each other. This has indeed been the subject of study in various contexts (Gamarnik, 2013; Gamarnik et al., 2014; Bamieh et al., 2002; Qu et al., 2019), where exponential decay properties or variants have been proposed and exploited for algorithmic design. In what follows, we provide the definition of the exponential decay property in the context of MARL and show its power in approximation of the exponentially large $Q$-function.

We define $N^{\kappa}_{-i} = \mathcal{N}/N^{\kappa}_i$, i.e. the set of agents that are outside of node $i$'s $\kappa$-hop neighborhood for some integer $\kappa$. We also write state $s$ as $(s_{N^{\kappa}_i}, s_{N^{\kappa}_{-i}})$, i.e. the states of agents that are in the $\kappa$-hop neighborhood of $i$ and outside of the $\kappa$-hop neighborhood respectively. The $(c, \rho)$ exponential decay property is defined below (Qu et al., 2019).

**Definition 1.** *The $(c, \rho)$ exponential decay property holds for $c > 0, \rho \in (0, 1)$, if for any $\theta$, for any $i \in \mathcal{N}$, $s_{N^{\kappa}_i} \in \mathcal{S}_{N^{\kappa}_i}$, $s_{N^{\kappa}_{-i}}, s'_{N^{\kappa}_{-i}} \in \mathcal{S}_{N^{\kappa}_{-i}}$, $a_{N^{\kappa}_i} \in \mathcal{A}_{N^{\kappa}_i}$, $a_{N^{\kappa}_{-i}}, a'_{N^{\kappa}_{-i}} \in \mathcal{A}_{N^{\kappa}_{-i}}$, the following holds,*

$$\left| Q^{\theta}_i(s_{N^{\kappa}_i}, s_{N^{\kappa}_{-i}}, a_{N^{\kappa}_i}, a_{N^{\kappa}_{-i}}) - Q^{\theta}_i(s_{N^{\kappa}_i}, s'_{N^{\kappa}_{-i}}, a_{N^{\kappa}_i}, a'_{N^{\kappa}_{-i}}) \right| \leq c\rho^{\kappa+1}. \tag{4}$$

The power of the exponential decay property is that it guarantees the dependence of $Q^{\theta}_i$ on other agents shrinks quickly as the distance between them grows. This naturally leads us to consider the following class of truncated $Q$-functions, where dependence on far-away nodes are "truncated",

$$\tilde{Q}^{\theta}_i(s_{N^{\kappa}_i}, a_{N^{\kappa}_i}) = \sum_{s_{N^{\kappa}_{-i}} \in \mathcal{S}_{N^{\kappa}_{-i}}, a_{N^{\kappa}_{-i}} \in \mathcal{A}_{N^{\kappa}_{-i}}} w_i(s_{N^{\kappa}_{-i}}, a_{N^{\kappa}_{-i}}; s_{N^{\kappa}_i}, a_{N^{\kappa}_i}) Q^{\theta}_i(s_{N^{\kappa}_i}, s_{N^{\kappa}_{-i}}, a_{N^{\kappa}_i}, a_{N^{\kappa}_{-i}}), \tag{5}$$

where $w_i(s_{N^{\kappa}_{-i}}, a_{N^{\kappa}_{-i}}; s_{N^{\kappa}_i}, a_{N^{\kappa}_i})$ are *any* non-negative weights satisfying

$$\sum_{s_{N^{\kappa}_{-i}} \in \mathcal{S}_{N^{\kappa}_{-i}}, a_{N^{\kappa}_{-i}} \in \mathcal{A}_{N^{\kappa}_{-i}}} w_i(s_{N^{\kappa}_{-i}}, a_{N^{\kappa}_{-i}}; s_{N^{\kappa}_i}, a_{N^{\kappa}_i}) = 1, \forall (s_{N^{\kappa}_i}, a_{N^{\kappa}_i}) \in \mathcal{S}_{N^k_i} \times \mathcal{A}_{N^k_i}. \tag{6}$$

The following lemma shows that the exponential decay property guarantees the truncated $Q$-function (5) is a good approximation of the full $Q$-function (3). Further, when using the truncated $Q$-functions in the policy gradient (Lemma 1) in place of the full $Q$ function, the exponential decay property enables an accurate approximation of the policy gradient, which is otherwise intractable to compute in its original form in Lemma 1. The proof of Lemma 2 can be found in Appendix A.1 in the supplementary material.

**Lemma 2.** *Under the $(c, \rho)$ exponential decay property, for any truncated Q-function in the form of (5), the following holds.*

(a) $\forall (s, a) \in \mathcal{S} \times \mathcal{A}$, $|Q^{\theta}_i(s, a) - \tilde{Q}^{\theta}_i(s_{N^{\kappa}_i}, a_{N^{\kappa}_i})| \leq c\rho^{\kappa+1}$.

(b) *Define the following approximated policy gradient,*

$$\hat{h}_i(\theta) = \mathbb{E}_{(s,a) \sim \pi^{\theta}} \left[ \frac{1}{n} \sum_{j \in N^{\kappa}_i} \tilde{Q}^{\theta}_j(s_{N^{\kappa}_j}, a_{N^{\kappa}_j}) \right] \nabla_{\theta_i} \log \zeta^{\theta_i}_i(a_i | s_i). \tag{7}$$

*Then, if $\|\nabla_{\theta_i} \log \zeta^{\theta_i}_i(a_i | s_i)\| \leq L_i$ for any $a_i$, $s_i$, we have $\|\hat{h}_i(\theta) - \nabla_{\theta_i} J(\theta)\| \leq cL_i\rho^{\kappa+1}$.*

The above results show the power of the exponential decay property – the $Q$ function and the policy gradient can be efficiently approximated despite their exponentially large dimension. These properties can be exploited to design scalable RL algorithms for networked systems.

While the exponential decay property is powerful, we have yet to show it actually holds under the average reward setting of this paper. In Qu et al. (2019), it was shown that the exponential decay property always holds when the objective is the discounted total reward. It turns out the average reward case is fundamentally different. In fact, it is not hard to encode NP-hard problems like graph coloring, 3-SAT into our setup (for an example, see Appendix A.2 in the supplementary material). As such, our problem is intractable in general, and there is no hope for the exponential decay property to hold generally. This is perhaps not surprising given the literature on MDPs with product state/action spaces (Blondel and Tsitsiklis, 2000; Papadimitriou and Tsitsiklis, 1999), which all point out the combinatorial nature and intractability of such multi-agent MDP problems.

Despite the worst case intractability, in the next section we show that a large class of subproblems in fact do satisfy the exponential decay property, for which we design a scalable RL algorithm utilizing the approximation guarantees in Lemma 2 implied by the exponential decay property.

## 3 Main Result

Despite the worst case intractability, in this section we identify a large subclass where learning is tractable, and illustrate this by developing a scalable RL algorithm for such problems. To this

end, in Section 3.1, we identify a general condition on the interaction among the agents under which the exponential decay property holds. Then, the exponential decay property, together with the idea of truncated $Q$-function and policy gradient outlined in Section 2.2, is combined with the actor critic framework in Section 3.2 to develop a Scalable Actor Critic algorithm that can find a $O(\rho^{\kappa+1})$-approximate stationary point of the objective function $J(\theta)$ with complexity scaling with the state-action space size of the largest $\kappa$-hop neighborhood.

## 3.1 Exponential Decay Holds when Interactions are Bounded

Our first result is our most technical and it identifies a rich class of problems when the exponential decay property holds.

**Theorem 1.** *Define*

$$
C_{ij} = \begin{cases} 0, & \text{if } j \notin N_i, \\ \sup_{s_{N_i/j},a_i} \sup_{s_j,s_j'} \mathrm{TV}(P_i(\cdot|s_j,s_{N_i/j},a_i), P_i(\cdot|s_j',s_{N_i/j},a_i)), & \text{if } j \in N_i, j \neq i, \\ \sup_{s_{N_i/i}} \sup_{s_i,s_i',a_i,a_i'} \mathrm{TV}(P_i(\cdot|s_i,s_{N_i/i},a_i), P_i(\cdot|s_i',s_{N_i/i},a_i')), & \text{if } j = i, \end{cases}
$$

*where* $\mathrm{TV}(\cdot,\cdot)$ *is the total variation distance bewteen two distributions. If for all* $i \in \mathcal{N}$, $\sum_{j=1}^n C_{ij} \leq \rho < 1$ *and* $|r_i(s_i,a_i)| \leq \bar{r}, \forall(s_i,a_i) \in \mathcal{S}_i \times \mathcal{A}_i$, *then the* $(\frac{\bar{r}}{1-\rho}, \rho)$ *exponential decay property holds.*

The proof of Theorem 1 can be found in Appendix B.1 in the supplementary material. The key analytic technique underlying our the proof is a novel MDP perturbation result (Lemma 3 in Appendix B.1 in the supplementary material). We show that, under the condition in Theorem 1, a perturbation on the state of an agent has exponential decaying affect on the state distribution of far away agents, which enables us to establish the exponential decay property.

In Theorem 1, $C_{ij}$ is the maximum possible change of the distribution of node $i$'next state as a result of a change in node $j$'th current state, where "change" is measured in total-variation distance. In other words, $C_{ij}$ can be interpreted as the strength of node $j$'s interaction with node $i$. With this interpretation, Theorem 1 shows that, for the exponential decay property to hold, we need (a) the interaction strength between each pair of nodes to be small enough; and (b) each node to have a small enough neighborhood. This is consistent with related conditions in the literature in exponential decay in static combinatorial optimizations, e.g. in Gamarnik (2013); Gamarnik et al. (2014), which requires the product between the maximum "interaction" and the maximum degree is bounded.

*Relation to discounted reward case.* In the $\gamma$-discounted total reward case in Qu et al. (2019), the exponential decay property is automatically true with decay rate $\gamma$ because for an agent to affect another far-way agent, it takes many time steps for the effect to take place, at which time the effect will have been significantly dampened as a result of the discounting factor. This stands in contrast with the average reward case, where it is unclear whether the exponential decay property holds or not as there is no discounting anymore. As a result, the analysis for the average reward turns out to be more difficult and requires very different techniques. Having said that, the result in the average reward case also has implications for the discounted reward case. Specifically, the results in Theorem 1 also leads to a sharper bound on the decay rate in the exponential decay property for the discounted reward case, showing the decay rate can be strictly smaller than the discounting factor $\gamma$. This is stated in the following corollary, and we provide a more detailed explanation on its setting and proof in Appendix B.3 in the supplementary material.

**Corollary 1.** *Under the conditions in Theorem 1, the* $(\frac{\bar{r}}{1-\gamma\rho}, \rho\gamma)$ *exponential decay property holds for the* $\gamma$-discounted reward case.

Finally, beyond the theoretic result, we also numerically verify that the exponential decay property holds widely for many randomly generated problem instances. We report these findings in Appendix B.4 in the supplementary material.

## 3.2 Scalable Actor-Critic Algorithm and Convergence Guarantee

Given the establishment of the exponential decay property in the previous section, a natural idea for algorithm design is to first estimate a truncated $Q$ function (5), compute the approximated policy gradient (7) and then do gradient step. In the following, we combine these ideas with the actor-critic framework in Konda and Tsitsiklis (2000) which uses Temporal Difference (TD) learning to

estimate the truncated $Q$-function, and present a Scalable Actor Critic Algorithm in Algorithm 1. In Algorithm 1, each step is conducted by all agents but is described from agent $i$'s perspective, and for the ease of presentation, we have used $z$ ($\mathcal{Z}$) to represent state-action pairs (spaces), e.g. $z_i(t) = (s_i(t), a_i(t)) \in \mathcal{Z}_i = \mathcal{S}_i \times \mathcal{A}_i$, $z_{N_i^\kappa}(t) = (s_{N_i^\kappa}(t), a_{N_i^\kappa}(t)) \in \mathcal{Z}_{N_i^\kappa} = \mathcal{S}_{N_i^\kappa} \times \mathcal{A}_{N_i^\kappa}$. The algorithm runs on a single trajectory and, at each iteration, it consists of two main steps, the critic (step 5 and 6) and the actor (step 7 and 8), which we describe in detail below.

- **The Critic.** In line with TD learning for average reward (Tsitsiklis and Van Roy, 1999), the critic consists of two variables, $\hat{\mu}_i^t$, whose purpose is to estimate the average reward; and $\hat{Q}_i^t$, the truncated $Q$ function, which follows the standard TD update. We note here that $\hat{Q}_i^t$ is one dimension less than the $\tilde{Q}_i$ defined in (5). In more detail, $\tilde{Q}_i \in \mathbb{R}^{\mathcal{Z}_{N_i^\kappa}}$ is defined on all state-action pairs in the neighborhood, but for $\hat{Q}_i^t$ we select an arbitrary dummy state-action pair $\tilde{z}_{N_i^\kappa} \in \mathcal{Z}_{N_i^\kappa}$, and $\hat{Q}_i^t \in \mathbb{R}^{\hat{\mathcal{Z}}_{N_i^\kappa}}$ is only defined on $\hat{\mathcal{Z}}_{N_i^\kappa} = \mathcal{Z}_{N_i^\kappa}/\{\tilde{z}_{N_i^\kappa}\}$. In other words, $\hat{Q}_i^t(z_{N_i^\kappa})$ is not defined for $z_{N_i^\kappa} = \tilde{z}_{N_i^\kappa}$, and when encountered, $\hat{Q}_i^t(\tilde{z}_{N_i^\kappa})$ is considered as 0. The reason for introducing such dummy state-action pair is mainly technical and is standard in value iteration for the average reward MDP, see e.g. Bertsekas (2007).

- **The Actor.** The actor computes the approximated gradient using the truncated $Q$-function according to Lemma 7 and follows a gradient step. Note that the step size in the gradient step contains a rescaling scalar factor $\Gamma(\hat{\mathbf{Q}}^t)$ which depends on the truncated $Q$-functions $\hat{\mathbf{Q}}^t = \{\hat{Q}_i^t\}_{i=1}^n$. Such a rescaling factor is used in Konda and Tsitsiklis (2000) and is mainly an artifact of the proof used to guarantee the actor does not move too fast when $\hat{Q}_i^t$ is large. In numerical experiments, we do not use this rescaling factor.

---

**Algorithm 1** Scalable Actor-Critic (SAC)

---

**Input:** $\theta_i(0)$; parameter $\kappa$; step size sequence $\{\alpha_t, \eta_t\}$.

1   Initialize $\hat{\mu}_i^0 = 0$ and $\hat{Q}_i^0$ to be the all zero vector in $\mathbb{R}^{\hat{\mathcal{Z}}_{N_i^\kappa}}$; start from random $z_i(0) = (s_i(0), a_i(0))$

2   **for** $t = 0, 1, 2, \ldots$ **do**

3      Receive reward $r_i(t) = r_i(s_i(t), a_i(t))$.

4      Get state $s_i(t+1)$, take action $a_i(t+1) \sim \zeta_i^{\theta_i(t)}(\cdot|s_i(t+1))$
     /* The critic.                                                               */

5      Update average reward $\hat{\mu}_i^{t+1} = (1 - \alpha_t)\hat{\mu}_i^t + \alpha_t r_i(t)$.

6      Update truncated $Q$ function. If $z_{N_i^\kappa}(t) = \tilde{z}_{N_i^\kappa}$, then set $\hat{Q}_i^{t+1}(z_{N_i^\kappa}) = \hat{Q}_i^t(z_{N_i^\kappa}), \forall z_{N_i^\kappa} \in \hat{\mathcal{Z}}_{N_i^\kappa}$.
     Otherwise, set

$$\hat{Q}_i^{t+1}(z_{N_i^\kappa}(t)) = (1 - \alpha_t)\hat{Q}_i^t(z_{N_i^\kappa}(t)) + \alpha_t(r_i(t) - \hat{\mu}_i^t + \hat{Q}_i^t(z_{N_i^\kappa}(t+1))),$$
$$\hat{Q}_i^{t+1}(z_{N_i^\kappa}) = \hat{Q}_i^t(z_{N_i^\kappa}), \quad \text{for } z_{N_i^\kappa} \in \hat{\mathcal{Z}}_{N_i^\kappa}/\{z_{N_i^\kappa}(t)\},$$

     with the understanding that $\hat{Q}_i^t(\tilde{z}_{N_i^\kappa}) = 0$.
     /* The actor.                                                              */

7      Calculate approximated gradient $\hat{g}_i(t) = \nabla_{\theta_i} \log \zeta_i^{\theta_i(t)}(a_i(t)|s_i(t))\frac{1}{n}\sum_{j \in N_i^\kappa} \hat{Q}_j^t(z_{N_j^\kappa}(t))$.

8      Conduct gradient step $\theta_i(t+1) = \theta_i(t) + \beta_t \hat{g}_i(t)$, where $\beta_t = \eta_t \Gamma(\hat{\mathbf{Q}}^t)$ and $\Gamma(\hat{\mathbf{Q}}^t) = \frac{1}{1 + \max_j \|\hat{Q}_j^t\|_\infty}$ is a rescaling scalar.

9   **end**

---

In the following, we prove a convergence guarantee for Scalable Acotr Critic method introduced above. To that end, we first describe the assumptions in our result. Our first assumption is that the rewards are bounded, a standard assumption in RL (Tsitsiklis and Van Roy, 1997).

**Assumption 1.** *For all $i$, and $s_i \in \mathcal{S}_i, a_i \in \mathcal{A}_i$, we have $0 \leq r_i(s_i, a_i) \leq \bar{r}$.*

Our next assumption is the exponential decay property, which as shown in Section 3.1, holds broadly for a large class of problems.

**Assumption 2.** *The $(c, \rho)$ exponential decay property holds.*

Our next assumption is on the step size and is standard (Konda and Tsitsiklis, 2000). We note that our algorithm uses two-time scales, meaning the actor progresses slower than the critic.

**Assumption 3.** *The positive step sizes $\alpha_t, \eta_t$ are deterministic, non-increasing, square summable but not summable[1] and satisfy $\sum_{t=0}^{\infty} (\frac{\eta_t}{\alpha_t})^d < \infty$ for some $d > 0$.*

Our next assumption is that the underlying problem is uniformly ergodic, which is again standard (Konda and Tsitsiklis, 2000).

**Assumption 4.** *Under any policy $\theta$, the induced Markov chain over the state-action space $\mathcal{Z} = \mathcal{S} \times \mathcal{A}$ is ergodic with stationary distribution $\pi^\theta$. Further, (a) For all $z \in \mathcal{Z}$, $\pi^\theta(z) \geq \sigma$ for some $\sigma > 0$. (b) $\|P^\theta - \mathbf{1}(\pi^\theta)^\top\|_{D^\theta} \leq \mu_D$ for some $\mu_D \in (0, 1)$, where $D^\theta = \mathrm{diag}(\pi^\theta) \in \mathbb{R}^{\mathcal{Z} \times \mathcal{Z}}$ and $\|\cdot\|_{D^\theta}$ is the weighted Euclidean norm $\|x\|_{D^\theta} = \sqrt{x^\top D^\theta x}$ for vectors $x \in \mathbb{R}^{\mathcal{Z}}$, and the corresponding induced norm for matrices.*

Note that, for Markov chains on the state-action pair that are ergodic, Assumption 4 are automatically true for some $\sigma > 0$ and $\mu_D \in (0, 1)$.[2] Assumption 4 also requires that it is true with constant $\sigma, \mu_D$ holds uniformly for all $\theta$. Such a uniform ergodicity assumption is common in the literature on actor-critic methods, e.g. Konda and Tsitsiklis (2003). Lastly, we assume the gradient is bounded and is Lipschitz continuous, again a standard assumption (Qu et al., 2019).

**Assumption 5.** *For each $i$, $a_i, s_i$, $\|\nabla_{\theta_i} \log \zeta_i^{\theta_i}(a_i|s_i)\| \leq L_i$. We let $L := \sqrt{\sum_{i \in \mathcal{N}} L_i^2}$. Further, $\nabla_{\theta_i} \log \zeta_i^{\theta_i}(a_i|s_i)$ is $L_i'$-Lipschitz continuous in $\theta_i$, and $\nabla J(\theta)$ is $L'$-Lipschitz continuous in $\theta$.*

We can now state our main convergence result. Note that the guarantee is asymptotic in nature. This is to be expected since, to the best of our knowledge, the finite time performance of two-time scale actor-critic algorithms on a single trajectory has long remained an open problem until the very recent progress of (Wu et al., 2020; Xu et al., 2020). We leave a finite time analysis of our algorithm as future work.

**Theorem 2.** *Under Assumptions 1-5, we have $\liminf_{T \to \infty} \|\nabla J(\theta(T))\| \leq L \frac{c\rho^{\kappa+1}}{1-\mu_D}$.*

Briefly speaking, Theorem 2 is a consequence of the power of exponential decay property in approximating the $Q$-function (Lemma 2), and its proof also uses tools from the stochastic approximation literature (Konda and Tsitsiklis, 2003). The complete proof of Theorem 2 is provided in Appendix D in the supplementary material. We comment that the asymptotic bound in Theorem 2 does not depend on parameters like $L', \bar{r}$, but these parameters will affect the convergence rate of the algorithm. From Theorem 2, our Scalable Actor Critic algorithm can find an approximated stationary point with gradient size $O(\rho^{\kappa+1})$ that decays exponentially in $\kappa$. Therefore, even for a small $\kappa$, our algorithm can find an approximate local minimizer in a *scalable* manner, with the complexity of the algorithm only scales with the state-action space size of the largest $\kappa$-hop neighborhood due to the use of the truncated $Q$ function, which could be much smaller than the full state-action space size (which are exponentially large in $n$) when the graph is sparse.

*Comparison of Theorem 2 to the discounted case.* Despite the similarity in the overall structure of the algorithms, the analysis of the algorithm in our case (Theorem 2) is very different from that in the discounted reward setting in Qu et al. (2019). Algorithmically, in Qu et al. (2019), at each time, there is a (long) inner loop of critic (TD-learning) steps that estimate the $Q$-function to a good accuracy during which the policy is *fixed*. This makes the analysis of the critic "decoupled" from the policy updates (actor). In our work, there is no inner loop, and the critic (TD-learning) and actor (policy update) steps are performed *simultaneously*. This creates many challenges in the analysis, as the critic no longer operates under a fixed policy, and the analysis of it cannot be decoupled from the actor. Therefore, our proof of Theorem 2 is drastically different from that in Qu et al. (2019) and requires tools from two-time-scale stochastic approximation (Konda and Tsitsiklis, 2003).

*Relationship between Theorem 1 and Theorem 2.* The conclusion of Theorem 1 is stated as Assumption 2 for Theorem 2. Other than that, Theorem 1 and Theorem 2 are independent results. On one hand, Theorem 1 focuses on the exponential decay property of $Q$ functions, which can be useful for

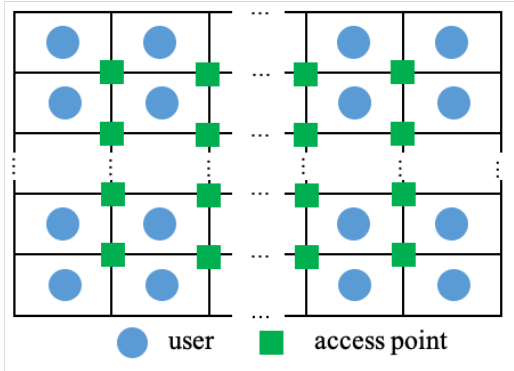

Figure 1: Setup of users and access points.

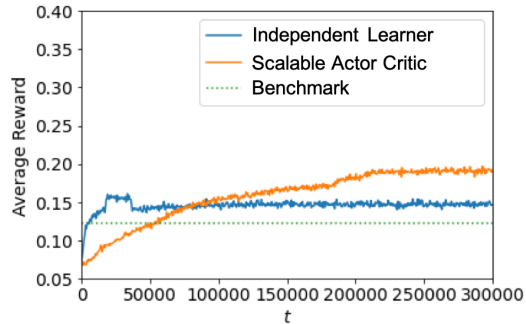

Figure 2: Average reward over the training process.

many types of RL methods beyond the actor-critic method studied in Theorem 2. On the other hand, the conditions in Theorem 1 might be conservative and the exponential decay may hold beyond the conditions in Theorem 1. Therefore, the conclusion of Theorem 2 may hold even if the conditions in Theorem 1 are not met.

## 4  Numerical Experiments

We consider a wireless network with multiple access points (Zocca, 2019; Lin et al., 2020), where there is a set of users $\mathcal{N} = \{u_1, u_2, \cdots, u_n\}$, and a set of network access points $Y = \{y_1, y_2, \cdots, y_m\}$. Each user $u_i$ only has access to a subset $Y_i \subseteq Y$ of the access points. We define the interaction graph as the conflict graph, in which two users $i$ and $j$ are neighbors if and only if they share an access point. Each user $i$ maintains a queue of packets defined as follows. At time step $t$, with probability $p_i$, user $i$ receives a new packet with an initial deadline $d_i$. Then, user $i$ can choose to send the earliest packet in its queue to one access point in its available set $Y_i$, or not sending at all. If an action of sending to $y_k \in Y_i$ is taken, and if no other users send to the same access point at this time, then the earliest packet in user $i$'s queue is transmitted with success probability $q_k$ which depends on the access point $y_k$; however, if another user also chooses to send to $y_k$, then there is a conflict and no transmission occurs. If the packet is successfully transmitted, it will be removed from user $i$'s queue and user $i$ will get a reward of 1. After this, the system moves to the next time step, with all deadlines of the remaining packets decreasing by 1 and packets with deadline 0 being discarded. In this example, the local state $s_i$ of user $i$ is a characterization of its queue of packets, and is represented by a $d_i$ binary tuple $s_i = (e_1, e_2, \cdots, e_{d_i}) \in \mathcal{S}_i = \{0, 1\}^{d_i}$, where for each $\ell \in \{1, \ldots, d_i\}$, $e_\ell \in \{0, 1\}$ indicates whether user $i$ has a packet with remaining deadline $\ell$. The action space is $\mathcal{A}_i = \{\text{null}\} \cup Y_i$, where null represents the action of not sending.

In the experiments, we set the deadline as $d_i = 2$, and all parameters $p_i$ (packet arrival probability for user $u_i$) and $q_k$ (success transmission probability for access point $y_k$) is generated uniformly random from $[0, 1]$. We consider a grid of $6 \times 6$ users in Figure 1, where each user has access points on the corners of its area. We run the Scalable Actor Critic algorithm with $\kappa = 1$ to learn a localized soft-max policy, starting from a initial policy where the action is chosen uniformly random. We compare the proposed method with a benchmark based on the localized ALOHA protocol (Roberts, 1975), where each user has a certain probability of sending the earliest packet and otherwise not sending at all. When it sends, it sends the packet to a random access point in its available set, with probability proportion to the success transmission probability of this access point and inverse proportion to the number of users that share this access point. We also compare our algorithm to the independent learner actor-critic algorithm Tan (1993); Lowe et al. (2017), where each agent runs a single-agent actor-critic algorithm using its local states and local actions. The results are shown in Figure 2. It shows that the proposed algorithm can outperform the ALOHA based benchmark, despite the proposed algorithm does not have access to the transmission probability $q_k$ which the benchmark has access to. The proposed approach also outperforms the independent learner algorithm, which is to be expected as the independent learner approach does not take into account the interactions from neighbors.

## Broader Impact

This paper contributes to the theoretical foundations of multi-agent reinforcement learning, with the goal of developing tools that can apply to the control of networked systems. The work can potentially lead to RL-based algorithms for the adaptive control of cyber-physical systems, such as the power grid, smart traffic systems, communication systems, and other smart infrastructure systems. While the approach is promising, as with other all theoretical work, it is limited by its assumptions. Applications of the proposed algorithm in its current form should be considered cautiously since the analysis here focuses on efficiency and does not consider the issue of fairness.

We see no ethical concerns related to this paper.

## Acknowledgements

We would like to thank Dr. Longbo Huang at Tsinghua University for suggesting the application example. The research was supported by Resnick Sustainability Institute Fellowship, NSF CAREER 1553407, ONR YIP, AFOSR YIP, the PIMCO Fellowship, and NSF grants AitF-1637598, CNS-1518941.

## Footnotes

[1] A sequence $\alpha_t$ is square summable if $\sum_{t=0}^{\infty} \alpha_t^2 < \infty$; is *not* summable if $\sum_{t=0}^{\infty} \alpha_t = \infty$.

[2] Assumption 4(b) is standard in the study of average reward TD learning, e.g. Tsitsiklis and Van Roy (1999, Sec. 4.3).

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
