[Supplementary Material]

## Supplementary Materials

## A  Appendix to Section 2.2

### A.1  Proof of Lemma 2

We first show part (a) that the truncated $Q$ function is a good approximation of the true $Q$ function. To see that, we have for any $(s, a) \in \mathcal{S} \times \mathcal{A}$, by (5) and (6),

$$|\tilde{Q}_i^\theta(s_{N_i^\kappa}, a_{N_i^\kappa}) - Q_i^\theta(s, a)|$$

$$= \Big| \sum_{s'_{N_{-i}^\kappa}, a'_{N_{-i}^\kappa}} w_i(s'_{N_{-i}^\kappa}, a'_{N_{-i}^\kappa}; s_{N_i^\kappa}, a_{N_i^\kappa}) Q_i^\theta(s_{N_i^\kappa}, s'_{N_{-i}^\kappa}, a_{N_i^\kappa}, a'_{N_{-i}^\kappa}) - Q_i^\theta(s_{N_i^\kappa}, s_{N_{-i}^\kappa}, a_{N_i^\kappa}, a_{N_{-i}^\kappa}) \Big|$$

$$\leq \sum_{s'_{N_{-i}^\kappa}, a'_{N_{-i}^\kappa}} w_i(s'_{N_{-i}^\kappa}, a'_{N_{-i}^\kappa}; s_{N_i^\kappa}, a_{N_i^\kappa}) \Big| Q_i^\theta(s_{N_i^\kappa}, s'_{N_{-i}^\kappa}, a_{N_i^\kappa}, a'_{N_{-i}^\kappa}) - Q_i^\theta(s_{N_i^\kappa}, s_{N_{-i}^\kappa}, a_{N_i^\kappa}, a_{N_{-i}^\kappa}) \Big|$$

$$\leq c\rho^{\kappa+1}, \tag{8}$$

where in the last step, we have used the $(c, \rho)$ exponential decay property, cf. Definition 1.

Next, we show part (b). Recall by the policy gradient theorem (Lemma 1),

$$\nabla_{\theta_i} J(\theta) = \mathbb{E}_{(s,a)\sim\pi^\theta} Q^\theta(s, a) \nabla_{\theta_i} \log \zeta^\theta(a|s) = \mathbb{E}_{(s,a)\sim\pi^\theta} Q^\theta(s, a) \nabla_{\theta_i} \log \zeta_i^{\theta_i}(a_i|s_i),$$

where we have used $\nabla_{\theta_i} \log \zeta^\theta(a|s) = \nabla_{\theta_i} \sum_{j \in \mathcal{N}} \log \zeta_j^{\theta_j}(a_j|s_j) = \nabla_{\theta_i} \log \zeta_i^{\theta_i}(a_i|s_i)$ by the localized policy structure. With the above equation, we can compute $\hat{h}_i(\theta) - \nabla_{\theta_i} J(\theta)$,

$$\hat{h}_i(\theta) - \nabla_{\theta_i} J(\theta) = \mathbb{E}_{(s,a)\sim\pi^\theta} \Big[ \frac{1}{n} \sum_{j \in N_i^\kappa} \tilde{Q}_j^\theta(s_{N_j^\kappa}, a_{N_j^\kappa}) - Q^\theta(s, a) \Big] \nabla_{\theta_i} \log \zeta_i^{\theta_i}(a_i|s_i)$$

$$= \mathbb{E}_{(s,a)\sim\pi^\theta} \Big[ \frac{1}{n} \sum_{j \in \mathcal{N}} \tilde{Q}_j^\theta(s_{N_j^\kappa}, a_{N_j^\kappa}) - \frac{1}{n} \sum_{j \in \mathcal{N}} Q_j^\theta(s, a) \Big] \nabla_{\theta_i} \log \zeta_i^{\theta_i}(a_i|s_i)$$

$$- \mathbb{E}_{(s,a)\sim\pi^\theta} \frac{1}{n} \sum_{j \in N_{-i}^\kappa} \tilde{Q}_j^\theta(s_{N_j^\kappa}, a_{N_j^\kappa}) \nabla_{\theta_i} \log \zeta_i^{\theta_i}(a_i|s_i)$$

$$:= E_1 - E_2.$$

We claim that $E_2 = 0$. To show this, note that $\pi^\theta(s, a) = d^\theta(s) \prod_{\ell=1}^n \zeta_\ell^{\theta_\ell}(a_\ell|s_\ell)$, where $d^\theta$ is the sationary distribution of the state. Then, for any $j \in N_{-i}^\kappa$, we have,

$$\mathbb{E}_{(s,a)\sim\pi^\theta} \nabla_{\theta_i} \log \zeta_i^{\theta_i}(a_i|s_i) \tilde{Q}_j^\theta(s_{N_j^\kappa}, a_{N_j^\kappa})$$

$$= \sum_{s,a} d^\theta(s) \prod_{\ell=1}^n \zeta_\ell^{\theta_\ell}(a_\ell|s_\ell) \frac{\nabla_{\theta_i} \zeta_i^{\theta_i}(a_i|s_i)}{\zeta_i^{\theta_i}(a_i|s_i)} \tilde{Q}_j^\theta(s_{N_j^\kappa}, a_{N_j^\kappa})$$

$$= \sum_{s,a} d^\theta(s) \prod_{\ell\neq i} \zeta_\ell^{\theta_\ell}(a_\ell|s_\ell) \nabla_{\theta_i} \zeta_i^{\theta_i}(a_i|s_i) \tilde{Q}_j^\theta(s_{N_j^\kappa}, a_{N_j^\kappa})$$

$$= \sum_{s,a_1,\ldots,a_{i-1},a_{i+1},\ldots,a_n} d^\theta(s) \prod_{\ell\neq i} \zeta_\ell^{\theta_\ell}(a_\ell|s_\ell) \tilde{Q}_j^\theta(s_{N_j^\kappa}, a_{N_j^\kappa}) \sum_{a_i} \nabla_{\theta_i} \zeta_i^{\theta_i}(a_i|s_i)$$

$$= 0, \tag{9}$$

where in the last equality, we have used $\tilde{Q}_j^\theta(s_{N_j^\kappa}, a_{N_j^\kappa})$ does not depend on $a_i$ as $i \notin N_j^\kappa$; and $\sum_{a_i} \nabla_{\theta_i} \zeta_i^{\theta_i}(a_i|s_i) = \nabla_{\theta_i} \sum_{a_i} \zeta_i^{\theta_i}(a_i|s_i) = \nabla_{\theta_i} 1 = 0$. Now that we have shown $E_2 = 0$, we can bound $E_1$ as follows,

$$\|\hat{h}_i(\theta) - \nabla_{\theta_i} J(\theta)\| = \|E_1\| \leq \mathbb{E}_{(s,a)\sim\pi^\theta} \frac{1}{n} \sum_{j \in \mathcal{N}} \Big| \tilde{Q}_j^\theta(s_{N_j^\kappa}, a_{N_j^\kappa}) - Q_j^\theta(s, a) \Big| \|\nabla_{\theta_i} \log \zeta_i^{\theta_i}(a_i|s_i)\|$$

$$\leq c\rho^{\kappa+1} L_i,$$

where in the last step, we have used (8) and the upper bound $\|\nabla_{\theta_i} \log \zeta_i^{\theta_i}(a_i|s_i)\| \leq L_i$. This concludes the proof of Lemma 2. $\square$

## A.2 An Example of Encoding NP-Hard Problems into MARL Setup

In this subsection, we provide an example on how NP hard problems can be encoded into the averge reward MARL problem with local interaction structure. We use the example of $k$-graph coloring in graph theory described as follows (Golovach et al., 2014). Given a graph $\mathcal{G} = (\mathcal{V}, \mathcal{E})$ and a set of $k$ colors $U$, a coloring is an assignment of a color $u \in U$ to each vertex in the graph, and a proper coloring is a coloring in which every two adjacent vertices have different colors. The $k$-graph coloring problem is to decide for a given graph, whether a proper coloring using $k$ colors exists, and is known to be NP hard when $k \geq 3$ (Golovach et al., 2014).

In what follows, we encode the $k$-coloring problem into our problem set up. Given a graph $\mathcal{G} = (\mathcal{V}, \mathcal{E})$, we identify the set of agents $\mathcal{N}$ with $\mathcal{V}$ and their interaction graph as $\mathcal{G}$. The local state is tuple $s_i = (u_i, b_i) \in \mathcal{S}_i = U \times \{0, 1\}$, where $u_i$ represents the color of node $i$ and $b_i$ is a binary variable indicating whether node $i$ has a different color from all nodes adjacent to $i$. We also identify action space $\mathcal{A}_i = U$ to be the set of colors. The state transition is given by the following rule, which satisfy the local interaction structure in (1): given $s_j(t) = (u_j(t), b_j(t))$ for $j \in N_i$ and $a_i(t)$, we set $u_i(t+1) = a_i(t)$, and if for all neighbors $j \in N_i/\{i\}$, $u_i(t)$ does not have the same color as $u_j(t)$, we set $b_i(t+1) = 1$; otherwise, set $b_i(t+1) = 0$.

Given $s_i = (u_i, b_i)$ and $a_i$, we also set the local reward $r_i(s_i, a_i) = 1$ if $b_i = 1$ (i.e. node $i$ does not have the same color as any of its adjacent nodes), and otherwise the reward is set as $0$. The local policy class is such that $a_i(t)$ is not allowed to depend on $s_i(t)$ but can be drawn from any distribution on the action space, i.e. the set of colors. In other words, $\zeta_i^{\theta_i}(\cdot)$ is a distribution on the action space, parameterized by $\theta_i$.

For policy $\theta = (\theta_i)_{i \in \mathcal{V}}$, it is clear that the stationary distribution of $s_i$ is simply $\zeta_i^{\theta_i}$; the stationary distribution for $b_i$, which we denote as $\pi_{b_i}^\theta$, is given by, $\pi_{b_i}^\theta(1) = \mathbb{P}(a_i \neq a_j, \forall j \in N_i/\{i\})$, where in the probability, $a_i$ is independently sampled from $\zeta_i^{\theta_i}$ and $a_j$ from $\zeta_j^{\theta_j}$. Further, in this case, the objective function (average reward) is given by

$$J(\theta) = \frac{1}{|\mathcal{V}|} \sum_{i \in \mathcal{V}} \pi_{b_i}^\theta(1).$$

It is immediately clear that in the above set up, the maximum possible average reward is $1$ if and only if there exists a proper coloring in the $k$-coloring problem. To see this, if there exists a proper coloring $(u_i^*)_{i \in \mathcal{V}}$ in the $k$-coloring problem, then a policy that always sets $a_i(t) = u_i^*$ will drive $s_i(t)$ in two steps to a fixed state $s_i = (u_i^*, 1)$, which will result in average reward $1$. On the contrary, if there exists a policy achieving average reward $1$, then the support of the action distribution in the policy constitute a set of proper colorings.

As such, if we can maximize the average reward, then we can also solve the $k$-coloring problem, which is known to be NP-hard when $k \geq 3$. This highlights the difficulty of the average reward MARL problem.

# B The exponential decay property and proof of Theorem 1

In this section, we formally prove Theorem 1 in Appendix B.1 that bounded interaction guarantees the exponential decay property holds. We will also provide a proof of Corollary 1 in Appendix B.3, and provide numerical validations of the exponential decay property in Appendix B.4.

## B.1 Proof of Theorem 1

Set $s = (s_{N_i^\kappa}, s_{N_{-i}^\kappa})$, $a = (a_{N_i^\kappa}, a_{N_{-i}^\kappa})$, and $\tilde{s} = (s_{N_i^\kappa}, \tilde{s}_{N_{-i}^\kappa})$, $\tilde{a} = (a_{N_i^\kappa}, \tilde{a}_{N_{-i}^\kappa})$. Recall the exponential decay property (Definition 1) is a bound on

$$\left| Q_i^\theta(s, a) - Q_i^\theta(\tilde{s}, \tilde{a}) \right|$$

$$\leq \sum_{t=0}^{\infty} \left| \mathbb{E}_\theta[r_i(s_i(t), a_i(t)) | s(0) = s, a(0) = a] - \mathbb{E}_\theta[r_i(s_i(t), a_i(t)) | s(0) = s', a(0) = a'] \right|$$

$$\leq \sum_{t=0}^{\infty} \mathrm{TV}(\pi_{t,i}^{\theta}, \tilde{\pi}_{t,i}^{\theta})\bar{r},$$

where $\pi_{t,i}^{\theta}$ means the distribution of $(s_i(t), a_i(t))$ conditioned on $(s(0), a(0)) = (s, a)$, and similarly $\tilde{\pi}_{t,i}^{\theta}$ is the distribution of $(s_i(t), a_i(t))$ conditioned on $(s(0), a(0)) = (\tilde{s}, \tilde{a})$. It is immediately clear that, $\pi_{t,i}^{\theta} = \tilde{\pi}_{t,i}^{\theta}$ for $t \leq \kappa$. Therefore, if we can show that

$$\mathrm{TV}(\pi_{t,i}^{\theta}, \tilde{\pi}_{t,i}^{\theta}) \leq \rho^t \text{ for } t > \kappa, \tag{10}$$

it immediately follows that

$$|Q_i^{\theta}(s, a) - Q_i^{\theta}(\tilde{s}, \tilde{a})| \leq \sum_{t=\kappa+1}^{\infty} \rho^t \bar{r} = \frac{\bar{r}}{1-\rho}\rho^{\kappa+1},$$

which is the desired exponential decay property.

We now show (10). Our primary tool is the following result on Markov chain with product state spaces, whose proof is deferred to Appendix B.2.

**Lemma 3.** *Consider a Markov Chain with state $z = (z_1, \ldots, z_n) \in \mathcal{Z} = \mathcal{Z}_1 \times \cdots \times \mathcal{Z}_n$, where each $\mathcal{Z}_i$ is some finite set. Suppose its transition probability factorizes as*

$$P(z(t+1)|z(t)) = \prod_{i=1}^{n} P_i(z_i(t+1)|z_{N_i}(t))$$

*and further, if $\sup_{1 \leq i \leq n} \sum_{j=1}^{n} C_{ij}^z \leq \rho$, where*

$$C_{ij}^z = \begin{cases} 0, & \text{if } j \notin N_i, \\ \sup_{z_{N_i/j}} \sup_{z_j, z_j'} \mathrm{TV}(P_i(\cdot|z_j, z_{N_i/j}), P_i(\cdot|z_j', z_{N_i/j})), & \text{if } j \in N_i, \end{cases}$$

*then for any $z = (z_{N_i^{\kappa}}, z_{N_{-i}^{\kappa}})$, $\tilde{z} = (z_{N_i^{\kappa}}, \tilde{z}_{N_{-i}^{\kappa}})$, we have,*

$$\mathrm{TV}(\pi_{t,i}, \tilde{\pi}_{t,i}) = 0 \quad \text{for} \quad t \leq \kappa, \qquad \mathrm{TV}(\pi_{t,i}, \tilde{\pi}_{t,i}) \leq \rho^t \quad \text{for} \quad t > \kappa,$$

*where $\pi_{t,i}$ is the distribution of $z_i(t)$ given $z(0) = z$, and $\tilde{\pi}_{t,i}$ is the distribution of $z_i(t)$ given $z(0) = \tilde{z}$.*

We now set the Markov chain in Lemma 3 to be the induced Markov chain of our MDP with a localized policy $\theta$, with $z_i = (s_i, a_i)$ and $\mathcal{Z}_i = \mathcal{S}_i \times \mathcal{A}_i$. For this induced chain, we have the transition factorized as,

$$P(s(t+1), a(t+1)|s(t), a(t)) = \prod_{i=1}^{n} \zeta_i^{\theta_i}(a_i(t+1)|s_i(t+1))P_i(s_i(t+1)|s_i(t), a_i(t), s_{N_i}(t)).$$

Then, $C_{ij}^z$ in Lemma 3 becomes

$$C_{ij}^z = \begin{cases} 0, & \text{if } j \notin N_i, \\ \sup_{s_{N_i/j}, a_i} \sup_{s_j, s_j'} \mathrm{TV}(P_i(\cdot|s_j, s_{N_i/j}, a_i), P_i(\cdot|s_j', s_{N_i/j}, a_i)), & \text{if } j \in N_i/i, \\ \sup_{s_{N_i/i}} \sup_{s_i, s_i', a_i, a_i'} \mathrm{TV}(P_i(\cdot|s_i, a_i, s_{N_i/i}), P_i(\cdot|s_i', a_i', s_{N_i/i})), & \text{if } j = i, \end{cases}$$

which is precisely the definition of $C_{ij}$ in Theorem 1. As a result, the condition in Theorem 1 implies the condition in Lemma 3 holds, regardless of the policy parameter $\theta$. Therefore, (10) holds and Theorem 1 is proven.

## B.2 Proof of Lemma 3

We do two runs of the Markov chain, one starting with $z$ with trajectory $z(0), \ldots, z(t), \ldots$, and another starting with $\tilde{z}$ with trajectory $\tilde{z}(0), \ldots, \tilde{z}(t), \ldots$ We use $\pi_t$ ($\tilde{\pi}_t$) to denote the distribution of $z(t)$ ($\tilde{z}(t)$); $\pi_{t,i}$ ($\tilde{\pi}_{t,i}$) to be the distribution of $z_i(t)$ ($\tilde{z}_i(t)$), $\pi_{t,N_i^{\kappa}}$ ($\tilde{\pi}_{t,N_i^{\kappa}}$) to denote the distribution of $z_{N_i^{\kappa}}(t)$ ($\tilde{z}_{N_i^{\kappa}}(t)$).

Our proof essentially relies on induction on $t$, and the following Lemma is the key step in the induction.

**Lemma 4.** *Given $t$, we say $a = [a_1, \ldots, a_n]^\top$ is $(t-1)$-compatible if for any $i, \kappa$, and for any function $f : \mathbb{R}^{\mathcal{Z}_{N_i^\kappa}} \to \mathbb{R}$,*

$$|\mathbb{E}_{z_{N_i^\kappa} \sim \pi_{t-1, N_i^\kappa}} f(z_{N_i^\kappa}) - \mathbb{E}_{z_{N_i^\kappa} \sim \tilde{\pi}_{t-1, N_i^\kappa}} f(z_{N_i^\kappa})| \leq \sum_{j \in N_i^\kappa} a_j \delta_j(f),$$

*where $\delta_j(f)$ is the variation of $f$'s dependence on $z_j$, i.e. $\delta_j(f) = \sup_{z_{N_i^\kappa/j}} \sup_{z_j, z_j'} |f(z_j, z_{N_i^\kappa/j}) - f(z_j', z_{N_i^\kappa/j})|$. Suppose now that $a$ is $(t-1)$-compatible, then we have $a' = [a_1', \ldots, a_n']^\top$ is $t$-compatible, with $a' = Ca$, where $C \in \mathbb{R}^{n \times n}$ is the matrix of $[C_{ij}^z]$.*

Now we use Lemma 4 to prove Lemma 3. We fix $i$ and $\kappa$. Since $z_{N_i^\kappa}(0) = \tilde{z}_{N_i^\kappa}(0)$, we can set $a_j^{(0)} = 0$ for $j \in N_i^\kappa$, and $a_j^{(0)} = 1$ for $j \notin N_i^\kappa$. It is easy to check such $a^{(0)}$ is 0-compatible. As a result, $a^{(t)} = C^t a^{(0)}$ is $t$-compatible. Since $a^{(0)}$ is supported outside $N_i^\kappa$, we have for all $t \leq \kappa$, $a_i^{(t)} = 0$; and for $t \geq \kappa + 1$, $a_i^{(t)} = [C^t a^{(0)}]_i \leq (\|C\|_\infty)^t \|a^{(0)}\|_\infty \leq \rho^t$. As a result, by the definition of $t$-compatible, we set $f : \mathbb{R}^{\mathcal{Z}_i} \to \mathbb{R}$ to be the indicator function for any event $A_i \subset \mathcal{Z}_i$ (i.e. $f(z_i) = \mathbf{1}(z_i \in A_i)$) and get,

$$|\pi_{t,i}(z_i \in A_i) - \tilde{\pi}_{t,i}(z_i \in A_i)| \leq a_i^{(t)},$$

and if we take the sup over $A_i$, we directly get,

$$\mathrm{TV}(\pi_{t,i}, \tilde{\pi}_{t,i}) \leq a_i^{(t)} \leq \rho^t,$$

which finishes the proof of Lemma 3. It remains to prove the induction step Lemma 4, which is done below.

*Proof of Lemma 4:* Recall that the transition probability can be factorized as follows,

$$P(z(t+1)|z(t)) = \prod_{i=1}^n P_i(z_i(t+1)|z_{N_i}(t)),$$

where the distribution of $z_i(t+1)$ only depends on $z_{N_i}(t)$ with transition probability given by $P_i(z_i(t+1)|z_{N_i}(t))$. We also define $P_{N_i^k}$ to be the transition from $z_{N_i^{k+1}}(t)$ to $z_{N_i^k}(t+1)$,

$$P_{N_i^k}(z_{N_i^k}(t+1)|z_{N_i^{k+1}}(t)) = \prod_{j \in N_i^k} P_j(z_j(t+1)|z_{N_j}(t)).$$

With these definitions, we have for any $i, \kappa$, and for any function $f : \mathbb{R}^{\mathcal{Z}_{N_i^\kappa}} \to \mathbb{R}$,

$$\left| \mathbb{E}_{z_{N_i^\kappa} \sim \pi_{t, N_i^\kappa}} f(z_{N_i^\kappa}) - \mathbb{E}_{z_{N_i^\kappa} \sim \tilde{\pi}_{t, N_i^\kappa}} f_{N_i^\kappa}(z_{N_i^\kappa}) \right| =$$

$$= \left| \mathbb{E}_{z'_{N_i^{\kappa+1}} \sim \pi_{t-1, N_i^{\kappa+1}}} \mathbb{E}_{z_{N_i^\kappa} \sim P_{N_i^\kappa}(\cdot|z'_{N_i^{\kappa+1}})} f(z_{N_i^\kappa}) - \mathbb{E}_{z'_{N_i^{\kappa+1}} \sim \tilde{\pi}_{t-1, N_i^{\kappa+1}}} \mathbb{E}_{z_{N_i^\kappa} \sim P_{N_i^\kappa}(\cdot|z'_{N_i^{\kappa+1}})} f(z_{N_i^\kappa}) \right|$$

$$= \left| \mathbb{E}_{z'_{N_i^{\kappa+1}} \sim \pi_{t-1, N_i^{\kappa+1}}} g(z'_{N_i^{\kappa+1}}) - \mathbb{E}_{z'_{N_i^{\kappa+1}} \sim \tilde{\pi}_{t-1, N_i^{\kappa+1}}} g(z'_{N_i^{\kappa+1}}) \right|, \tag{11}$$

where we have defined $g(z'_{N_i^{\kappa+1}}) = \mathbb{E}_{z_{N_i^\kappa} \sim P_{N_i^\kappa}(\cdot|z'_{N_i^{\kappa+1}})} f(z_{N_i^\kappa})$. Since $a = [a_1, \ldots, a_n]^\top$ is $(t-1)$ compatible, we have,

$$\left| \mathbb{E}_{z'_{N_i^{\kappa+1}} \sim \pi_{t-1, N_i^{\kappa+1}}} g(z'_{N_i^{\kappa+1}}) - \mathbb{E}_{z'_{N_i^{\kappa+1}} \sim \tilde{\pi}_{t-1, N_i^{\kappa+1}}} g(z'_{N_i^{\kappa+1}}) \right| \leq \sum_{j \in N_i^{\kappa+1}} a_j \delta_j(g).$$

Now we analyze $\delta_j(g)$. We fix $z'_{N_i^{\kappa+1}/j}$, then

$$g(z_j', z'_{N_i^{\kappa+1}/j}) - g(z_j'', z'_{N_i^{\kappa+1}/j}) = \mathbb{E}_{z_{N_i^\kappa} \sim P_{N_i^\kappa}(\cdot|z_j', z'_{N_i^{\kappa+1}/j})} f(z_{N_i^\kappa}) - \mathbb{E}_{z_{N_i^\kappa} \sim P_{N_i^\kappa}(\cdot|z_j'', z'_{N_i^{\kappa+1}/j})} f(z_{N_i^\kappa}).$$

Taking a closer look, both $P_{N_i^\kappa}(\cdot|z_j', z'_{N_i^{\kappa+1}/j})$ and $P_{N_i^\kappa}(\cdot|z_j'', z'_{N_i^{\kappa+1}/j})$ are product distributions on the states in $N_i^\kappa$, and they differ only for those $\ell \in N_i^\kappa$ that are adjacent to $j$, i.e. $N_i^\kappa \cap N_j$. Therefore, we can use the following auxiliary result whose proof is provided in the bottom of this subsection.

**Lemma 5.** *For a function $f$ that depends on a group of variables $z = (z_i)_{i \in V}$, let $P_i$ and $\tilde{P}_i$ to be two distributions on $z_i$. Let $P$ be the product distribution of $P_i$ and $\tilde{P}$ be the product distribution of $\tilde{P}_i$. Then*

$$|\mathbb{E}_{z \sim P} f(z) - \mathbb{E}_{z \sim \tilde{P}} f(z)| \leq \sum_{i \in V} \mathrm{TV}(P_i, \tilde{P}_i) \delta_i(f).$$

By Lemma 5, we have,

$$|g(z_j', z_{N_i^{\kappa+1}/j}') - g(z_j'', z_{N_i^{\kappa+1}/j}')| \leq \sum_{\ell \in N_i^\kappa \cap N_j} \mathrm{TV}(P_\ell(\cdot | z_j', z_{N_\ell/j}'), P_\ell(\cdot | z_j'', z_{N_\ell/j}')) \delta_\ell(f)$$

$$\leq \sum_{\ell \in N_i^\kappa \cap N_j} C_{\ell j}^z \delta_\ell(f).$$

As such, $\delta_j(g) \leq \sum_{\ell \in N_i^\kappa \cap N_j} C_{\ell j}^z \delta_\ell(f)$, and we can continue (11) and get,

$$\left| \mathbb{E}_{z_{N_i^\kappa} \sim \pi_{t, N_i^\kappa}} f(z_{N_i^\kappa}) - \mathbb{E}_{z_{N_i^\kappa} \sim \tilde{\pi}_{t, N_i^\kappa}} f_{N_i^\kappa}(z_{N_i^\kappa}) \right| \leq \sum_{j \in N_i^{\kappa+1}} a_j \delta_j(g)$$

$$\leq \sum_{j \in N_i^{\kappa+1}} a_j \sum_{\ell \in N_i^\kappa \cap N_j} C_{\ell j}^z \delta_\ell(f)$$

$$= \sum_{\ell \in N_i^\kappa} \sum_{j \in N_\ell} a_j C_{\ell j}^z \delta_\ell(f).$$

This implies $a' = [a_1', \ldots, a_n']^\top$ is $t$-compatible, where $a_\ell' = \sum_{j \in N_\ell} C_{\ell j}^z a_j$. $\qquad\square$

Finally, we provide the proof of Lemma 5.

*Proof of Lemma 5:* We do induction on the size of $|V|$. For $|V| = 1$, we have

$$|\mathbb{E}_{z_1 \sim P_1} f(z_1) - \mathbb{E}_{z_1 \sim \tilde{P}_1} f(z_1)| = |\langle P_1, f \rangle - \langle \tilde{P}_1, f \rangle|,$$

where both $P_1, \tilde{P}_1$ and $f$ are interpreted as vectors indexed by $z_1$, and $\langle \cdot, \cdot \rangle$ is the usual inner product. Let $\mathbf{1}$ be the all one vector with the same dimension of $P_1, \tilde{P}_1$ and $f$. Let $m$ and $M$ be the minimum and maximum value of $f$ respectively. Then,

$$|\langle P_1, f \rangle - \langle \tilde{P}_1, f \rangle| = |\langle P_1 - \tilde{P}_1, f - \frac{M+m}{2}\mathbf{1} \rangle|$$

$$\leq \|P_1 - \tilde{P}_1\|_1 \|f - \frac{M+m}{2}\mathbf{1}\|_\infty = \frac{M-m}{2} \|P_1 - \tilde{P}_1\|_1 = \mathrm{TV}(P_1, \tilde{P}_1) \delta_1(f).$$

As a result, the statement is true for $|V| = 1$. Suppose the statement is true for $|V| = n - 1$. Then, for $|V| = n$, we use $z_{2:n}$ to denote $(z_2, \ldots, z_n)$ and use $P_{2:n}$ to denote the product distribution $P_{2:n}(z_2, \ldots, z_n) = \prod_{i=2}^n P_i(z_i)$; $\tilde{P}_{2:n}$ is defined similarly. Then,

$$\mathbb{E}_{z \sim P} f(z) - \mathbb{E}_{z \sim \tilde{P}} f(z)| = |\mathbb{E}_{z_1 \sim P_1} \mathbb{E}_{z_{2:n} \sim P_{2:n}} f(z_1, z_{2:n}) - \mathbb{E}_{z_1 \sim \tilde{P}_1} \mathbb{E}_{z_{2:n} \sim \tilde{P}_{2:n}} f(z_1, z_{2:n})|$$

$$\leq |\mathbb{E}_{z_1 \sim P_1} \mathbb{E}_{z_{2:n} \sim P_{2:n}} f(z_1, z_{2:n}) - \mathbb{E}_{z_1 \sim P_1} \mathbb{E}_{z_{2:n} \sim \tilde{P}_{2:n}} f(z_1, z_{2:n})|$$

$$+ |\mathbb{E}_{z_1 \sim P_1} \mathbb{E}_{z_{2:n} \sim \tilde{P}_{2:n}} f(z_1, z_{2:n}) - \mathbb{E}_{z_1 \sim \tilde{P}_1} \mathbb{E}_{z_{2:n} \sim \tilde{P}_{2:n}} f(z_1, z_{2:n})|$$

$$\leq \mathbb{E}_{z_1 \sim P_1} |\mathbb{E}_{z_{2:n} \sim P_{2:n}} f(z_1, z_{2:n}) - \mathbb{E}_{z_{2:n} \sim \tilde{P}_{2:n}} f(z_1, z_{2:n})|$$

$$+ |\mathbb{E}_{z_1 \sim P_1} \bar{f}(z_1) - \mathbb{E}_{z_1 \sim \tilde{P}_1} \bar{f}(z_1)|,$$

where we have defined $\bar{f}(z_1) = \mathbb{E}_{z_{2:n} \sim \tilde{P}_{2:n}} f(z_1, z_{2:n})$. Fixing $z_1$, we have by induction assumption,

$$|\mathbb{E}_{z_{2:n} \sim P_{2:n}} f(z_1, z_{2:n}) - \mathbb{E}_{z_{2:n} \sim \tilde{P}_{2:n}} f(z_1, z_{2:n})| \leq \sum_{i=2}^n \mathrm{TV}(P_i, \tilde{P}_i) \delta_i(f(z_1, \cdot)) \leq \sum_{i=2}^n \mathrm{TV}(P_i, \tilde{P}_i) \delta_i(f).$$

Further, we have,

$$\delta_1(\bar{f}) = \sup_{z_1, z_1'} |\mathbb{E}_{z_{2:n} \sim \tilde{P}_{2:n}} f(z_1, z_{2:n}) - \mathbb{E}_{z_{2:n} \sim \tilde{P}_{2:n}} f(z_1', z_{2:n})|$$

$$\leq \sup_{z_1, z_1'} \mathbb{E}_{z_{2:n} \sim \tilde{P}_{2:n}} |f(z_1, z_{2:n}) - f(z_1', z_{2:n})|$$

$$\leq \sup_{z_1, z_1'} \sup_{z_{2:n}} |f(z_1, z_{2:n}) - f(z_1', z_{2:n})| = \delta_1(f).$$

Combining these results, we have

$$|\mathbb{E}_{s \sim P} f(s) - \mathbb{E}_{s \sim \tilde{P}} f(s)| \leq \mathbb{E}_{z_1 \sim P_1} \sum_{i=2}^{n} \mathrm{TV}(P_i, \tilde{P}_i) \delta_i(f) + \mathrm{TV}(P_1, \tilde{P}_1) \delta_1(\bar{f})$$

$$\leq \sum_{i=1}^{n} \mathrm{TV}(P_i, \tilde{P}_i) \delta_i(f).$$

So the induction is finished and the proof of Lemma 5 is concluded. $\qquad\square$

### B.3 Proof of Corollary 1

In the $\gamma$-discounted case, the $Q$-function is defined as (Qu et al., 2019),

$$Q_i^\theta(s, a) = \mathbb{E}_{a(t) \sim \zeta^\theta(\cdot|s(t))} \left[ \sum_{t=0}^{\infty} \gamma^t r_i(s_i(t), a_i(t)) \bigg| s(0) = s, a(0) = a \right]. \tag{12}$$

For notational simplicity, denote $s = (s_{N_i^\kappa}, s_{N_{-i}^\kappa})$, $a = (a_{N_i^\kappa}, a_{N_{-i}^\kappa})$; $s' = (s_{N_i^\kappa}, s'_{N_{-i}^\kappa})$ and $a' = (a_{N_i^\kappa}, a'_{N_{-i}^\kappa})$. Let $\pi_{t,i}$ be the distribution of $(s_i(t), a_i(t))$ conditioned on $(s(0), a(0)) = (s, a)$ under policy $\theta$, and let $\pi'_{t,i}$ be the distribution of $(s_i(t), a_i(t))$ conditioned on $(s(0), a(0)) = (s', a')$ under policy $\theta$. Then, under the conditions of Theorem 1, we can use equation (10) in the proof of Theorem 1 (also see Lemma 3), which still holds in the discounted setting as equation (10) is a property of the underlying Markov chain, irrespective of how the objective is defined. This leads to,

$$\mathrm{TV}(\pi_{t,i}, \pi'_{t,i}) = 0 \quad \text{for} \quad t \leq \kappa, \qquad \mathrm{TV}(\pi_{t,i}, \pi'_{t,i}) \leq \rho^t \quad \text{for} \quad t > \kappa.$$

With these preparations, we verify the exponential decay property. We expand the definition of $Q_i^\theta$ in (12),

$$|Q_i^\theta(s, a) - Q_i^\theta(s', a')|$$

$$\leq \sum_{t=0}^{\infty} \left| \mathbb{E}\big[\gamma^t r_i(s_i(t), a_i(t))\big|(s(0), a(0)) = (s, a)\big] - \mathbb{E}\big[\gamma^t r_i(s_i(t), a_i(t))\big|(s(0), a(0)) = (s', a')\big] \right|$$

$$= \sum_{t=0}^{\infty} \left| \gamma^t \mathbb{E}_{(s_i, a_i) \sim \pi_{t,i}} r_i(s_i, a_i) - \gamma^t \mathbb{E}_{(s_i, a_i) \sim \pi'_{t,i}} r_i(s_i, a_i) \right|$$

$$\leq \sum_{t=0}^{\infty} \gamma^t \bar{r} \mathrm{TV}(\pi_{t,i}, \pi'_{t,i}) \leq \sum_{t=\kappa+1}^{\infty} \gamma^t \bar{r} \rho^t \leq \frac{\bar{r}}{1 - \gamma\rho} (\gamma\rho)^{\kappa+1}.$$

The above inequality shows that the $(\frac{\bar{r}}{1-\rho\gamma}, \rho\gamma)$-exponential decay property holds and concludes the proof of Corollary 1.

### B.4 Numerical Validation of the Exponential Decay Property

In this subsection, we conduct numerical experiments to show that the exponential decay property holds broadly for randomly generated problem instances.

We consider a line graph with $n = 10$ nodes, local state space size $|\mathcal{S}_i| = 2$, local action space size $|\mathcal{A}_i| = 3$. We generate the local transition probabilities $P_i$, localized polices $\zeta_i$ and local rewards $r_i$ uniformly randomly with maximum reward set to be 1. To verify the exponential decay property, we consider Definition 1, where we pick $i$ to be the left most node in the line, generate $s, s', a, a'$ uniformly random in the global state or action space, and then increase $\kappa$ from 0 to $n - 2$. For each $\kappa$, we calculate the left hand side of (4) exactly through brutal force. We repeat the above procedure

Figure 3: Numerical verification of the exponential decay property. The $y$-axis is the left hand side of (4) whereas the $x$-axis is $\kappa$. The solid line represents the median value of different runs, whereas the shaded region represents 10% to 90% percentile of the runs.

100 times, each time with a newly generated instance, and plot the left hand side of (4) as a function of $\kappa$ in Figure 3a.

We do a similar experiment on a $2 \times 6$ 2D grid, with a similar setup except node $i$ is now selected as the corner node in the grid. The results are shown in Figure 3b. Both Figure 3a and Figure 3b confirm that the left hand side of (4) decay exponentially in $\kappa$. This shows that the exponential decay property holds broadly for instances generated randomly.

## C   Analysis of the Critic

The goal of the section is to analyze the critic update (line 5 and 6 in Algorithm 1). Our algorithm is a two-time scale algorithm, where the critic runs faster than the actor policy parameter $\theta(t)$. Therefore, in what follows, we show that the truncated $Q$-function in the critic $\hat{Q}_i^t$ "tracks" a quantity $\hat{Q}_i^{\theta(t)}$, which is the fixed point of the critic update when the policy is "frozen" at $\theta(t)$. Further, we show that this fixed point is a good approximation of the true $Q$ function $Q_i^{\theta(t)}$ for policy $\theta(t)$ because of the exponential decay property. The formal statement is given in Theorem 3.

**Theorem 3.** *The following two statements are true.*

(a) *For each $i$ and $\theta$, there exists $\hat{Q}_i^\theta \in \mathbb{R}^{\hat{\mathcal{Z}}_{N_i^\kappa}}$ which is an approximation of the true Q function in the sense that, there exists scalar $c_i^\theta$ that depends on $\theta$, such that*

$$\sqrt{\mathbb{E}_{z \sim \pi^\theta} |\hat{Q}_i^\theta(z_{N_i^\kappa}) + c_i^\theta - Q_i^\theta(z)|^2} \leq \frac{c\rho^{\kappa+1}}{1-\mu_D}, \tag{13}$$

*where $\hat{Q}_i^\theta(\tilde{z}_{N_i^\kappa})$ is understood as 0.*

(b) *For each $i$, almost surely $\sup_{t \geq 0} \|\hat{Q}_i^t\|_\infty < \infty$. Further, $\hat{Q}_i^t$ tracks $\hat{Q}_i^{\theta(t)}$ in the sense that almost surely, $\lim_{t \to \infty} \hat{Q}_i^t - \hat{Q}_i^{\bar{\theta}(t)} = 0$.*

Our proof relies on the result on two-time scale stochastic approximation in Konda and Tsitsiklis (2003). In Appendix C.1, we review the result in Konda and Tsitsiklis (2003) and in Appendix C.2, we provide the proof for Theorem 3.

### C.1   Review of A Stochastic Approximation Result

In this subsection, we review a result on two time-scale stochastic approximation in Konda and Tsitsiklis (2003) which will be used in our proof for Theorem 3. Consider the following iterative

stochastic approximation scheme with iterate $x^t \in \mathbb{R}^m$,[3]

$$x^{t+1} = x^t + \alpha_t(h^{\theta(t)}(z(t)) - G^{\theta(t)}(z(t))x^t + \xi^{t+1}x^t), \qquad (14a)$$

$$\theta(t+1) = \theta(t) + \eta_t H^{t+1}, \qquad (14b)$$

where $z(t)$ is a stochastic process with finite state space $\mathcal{Z}$; $h^\theta(\cdot) : \mathcal{Z} \to \mathbb{R}^m, G^\theta(\cdot) : \mathcal{Z} \to \mathbb{R}^{m \times m}$ are vectors or matrices depending on both parameter $\theta$ as well as the state $z$; $\xi^{t+1} \in \mathbb{R}^{m \times m}$ and $H^{t+1}$ is some vector that drives the change of $\theta(t)$.

In what follows, we state Assumption 6 to 11 used in Konda and Tsitsiklis (2003). Assumption 6 is related to the summability of the step size $\alpha_t$.

**Assumption 6.** *The step size is deterministic, nonincreasing, and satisfies $\sum_t \alpha_t = \infty, \sum_t \alpha_t^2 < \infty$.*

Let $\mathcal{F}_t$ be the $\sigma$ algebra generated by $\{z(k), H^k, x^k, \theta(k)\}_{k \leq t}$. Assumption 7 says that the stochastic process $z(t)$ is Markovian and is driven by a transition kernal that depends on $\theta(t)$.

**Assumption 7.** *There exists a parameterized family of transition kernels $P^\theta$ on state space $\mathcal{Z}$ such that, for every $A \subset \mathcal{Z}$, $\mathbb{P}(z(t+1) \in A | \mathcal{F}_t) = \mathbb{P}(z(t+1) \in A | z(t), \theta(t)) = P^{\theta(t)}(z(t+1) \in \mathcal{A} | z(t))$.*

Assumption 8 is a technical assumption on the transition kernel $P^\theta$ as well as $h^\theta, G^\theta$.

**Assumption 8.** *For each $\theta$, there exists function $\bar{h}(\theta) \in \mathbb{R}^m, \bar{G}(\theta) \in \mathbb{R}^{m \times m}, \hat{h}^\theta : \mathcal{Z} \to \mathbb{R}^m, \hat{G}^\theta : \mathcal{Z} \to \mathbb{R}^{m \times m}$ that satisfy the following.*

(a) *For all $z \in \mathcal{Z}$,*

$$\hat{h}^\theta(z) = h^\theta(z) - \bar{h}(\theta) + [P^\theta \hat{h}^\theta](z),$$

$$\hat{G}^\theta(z) = G^\theta(z) - \bar{G}(\theta) + [P^\theta \hat{G}^\theta](z),$$

*where $P^\theta \hat{h}^\theta$ is a map from $\mathcal{Z}$ to $\mathbb{R}^m$ given by $[P^\theta \hat{h}^\theta](z) = \mathbb{E}_{z' \sim P^\theta(\cdot | z)} \hat{h}^\theta(z')$; similarly, $P^\theta \hat{G}^\theta$ is given by $[P^\theta \hat{G}^\theta](z) = \mathbb{E}_{z' \sim P^\theta(\cdot | z)} \hat{G}^\theta(z')$.*

(b) *For some constant $C$, $\max(\|\bar{h}(\theta)\|, \|\bar{G}(\theta)\|) \leq C$ for all $\theta$.*

(c) *For any $d > 0$, there exists $C_d > 0$ such that $\sup_t \mathbb{E}\|f^{\theta(t)}(z(t))\|^d \leq C_d$ where $f^\theta$ represents any of the functions $\hat{h}^\theta, h^\theta, \hat{G}^\theta, G^\theta$.*

(d) *For some constant $C > 0$ and for all $\theta, \bar{\theta}$,*

$$\max(\|\bar{h}(\theta) - \bar{h}(\bar{\theta})\|, \|\bar{G}(\theta) - \bar{G}(\bar{\theta})\|) \leq C\|\theta - \bar{\theta}\|.$$

(e) *There exists a positive constant $C$ such that for each $z \in \mathcal{Z}$,*

$$\|P^\theta f^\theta(z) - P^{\bar{\theta}} f^{\bar{\theta}}(z)\| \leq C\|\theta - \bar{\theta}\|,$$

*where $f^\theta$ is any of the function $\hat{h}^\theta$ and $\hat{G}^\theta$.*

The next Assumption 9 is to ensure that $\theta(t)$ changes slowly by imposing a bound on $H^t$ and requiring step size $\eta_t$ to be much smaller than $\alpha_t$.

**Assumption 9.** *The process $H^t$ satisfies $\sup_t \mathbb{E}|H^t|^d < \infty$ for all $d$. Further, the sequence $\eta_t$ is deterministic and satisfies $\sum_t \left(\frac{\eta_t}{\alpha_t}\right)^d < \infty$ for some $d > 0$.*

Assumption 10 says that the $\xi^t$ is a martingale difference sequence.

**Assumption 10.** *$\xi^t$ is an $m \times m$ matrix valued $\mathcal{F}_t$-martingale difference, with bounded momemnts, i.e.*

$$\mathbb{E}\xi^{t+1}|\mathcal{F}_t = 0, \quad \sup_t \mathbb{E}\|\xi^{t+1}\|^d < \infty,$$

*for each $d > 0$.*

The final Assumption 11 requires matrix $\bar{G}(\theta)$ to be uniformly positive definite.

**Assumption 11** (Uniform Positive Definiteness). *There exists $a > 0$ s.t. for all $x \in \mathbb{R}^m$ and $\theta$, we have*

$$x^\top \bar{G}(\theta) x \geq a\|x\|^2.$$

With the above assumtions, Konda and Tsitsiklis (2003, Lem. 12, Thm. 7) shows that the following theorem holds.

**Theorem 4** (Konda and Tsitsiklis (2003)). *Under Assumption 6-11, with probability* 1, $\sup_{t\geq 0} \|x^t\| < \infty$ *and*

$$\lim_{t\to\infty} \|x^t - \bar{G}(\theta(t))^{-1}\bar{h}(\theta(t))\| = 0.$$

In the next subsection, we will use the stochastic approximation result here to provide a proof of Theorem 3.

## C.2 Proof of Theorem 3

In this subsection, we will write our algorithm in the form of the stochastic approximation scheme (14) and provide a proof of Theorem 3. Throughout the rest of the section, we fix $i \in \mathcal{N}$.

Define $\mathbf{e}_{z_{N_i^\kappa}} \in \mathbb{R}^{\hat{\mathcal{Z}}_{N_i^\kappa}}$ to be the unit vector in $\mathbb{R}^{\hat{\mathcal{Z}}_{N_i^\kappa}}$ when $z_{N_i^\kappa} \neq \tilde{z}_{N_i^\kappa}$, and is the zero vector when $z_{N_i^\kappa} = \tilde{z}_{N_i^\kappa}$ (the dummy state-action pair). Then, one can check that the critic part of our algorithm (line 5 and 6 in Algorithm 1) can be rewritten as,

$$\hat{\mu}_i^{t+1} = \hat{\mu}_i^t + \alpha_t[r_i(z_i(t)) - \hat{\mu}_i^t], \tag{15}$$

$$\hat{Q}_i^{t+1} = \hat{Q}_i^t + \alpha_t[r_i(z_i(t)) - \hat{\mu}_i^t + \mathbf{e}_{z_{N_i^\kappa}(t+1)}^\top \hat{Q}_i^t - \mathbf{e}_{z_{N_i^\kappa}(t)}^\top \hat{Q}_i^t]\mathbf{e}_{z_{N_i^\kappa}(t)}. \tag{16}$$

When written in vector form, the above equation becomes

$$\begin{bmatrix} \hat{\mu}_i^{t+1} \\ \hat{Q}_i^{t+1} \end{bmatrix} = \begin{bmatrix} \hat{\mu}_i^t \\ \hat{Q}_i^t \end{bmatrix} + \alpha_t\left[ - \begin{bmatrix} 1 & 0 \\ \mathbf{e}_{z_{N_i^\kappa}(t)} & \mathbf{e}_{z_{N_i^\kappa}(t)}[\mathbf{e}_{z_{N_i^\kappa}(t)}^\top - \mathbf{e}_{z_{N_i^\kappa}(t+1)}^\top] \end{bmatrix} \begin{bmatrix} \hat{\mu}_i^t \\ \hat{Q}_i^t \end{bmatrix} + \begin{bmatrix} r_i(z_i(t)) \\ \mathbf{e}_{z_{N_i^\kappa}(t)}r_i(z_i(t)) \end{bmatrix} \right].$$

We rescale the $\hat{\mu}_i^t$ coordinate by a factor of $c'$ for technical reasons to be clear later, and rewrite the above equation in an equivalent form,

$$\begin{bmatrix} c'\hat{\mu}_i^{t+1} \\ \hat{Q}_i^{t+1} \end{bmatrix} = \begin{bmatrix} c'\hat{\mu}_i^t \\ \hat{Q}_i^t \end{bmatrix} + \alpha_t\left[ - \begin{bmatrix} 1 & 0 \\ \frac{1}{c'}\mathbf{e}_{z_{N_i^\kappa}(t)} & \mathbf{e}_{z_{N_i^\kappa}(t)}[\mathbf{e}_{z_{N_i^\kappa}(t)}^\top - \mathbf{e}_{z_{N_i^\kappa}(t+1)}^\top] \end{bmatrix} \begin{bmatrix} c'\hat{\mu}_i^t \\ \hat{Q}_i^t \end{bmatrix} + \begin{bmatrix} c'r_i(z_i(t)) \\ \mathbf{e}_{z_{N_i^\kappa}(t)}r_i(z_i(t)) \end{bmatrix} \right].$$

We define $x_i^t = [c'\hat{\mu}_i^t; \hat{Q}_i^t]$ and,

$$\tilde{G}_i(z, z') = \begin{bmatrix} 1 & 0 \\ \frac{1}{c'}\mathbf{e}_{z_{N_i^\kappa}} & \mathbf{e}_{z_{N_i^\kappa}}[\mathbf{e}_{z_{N_i^\kappa}}^\top - \mathbf{e}_{z'_{N_i^\kappa}}^\top] \end{bmatrix}, \quad h_i(z) = \begin{bmatrix} c'r_i(z_i) \\ \mathbf{e}_{z_{N_i^\kappa}}r_i(z_i) \end{bmatrix}.$$

With the above definitions, the critic update equation (15) and (16) can be rewritten as the following,

$$x_i^{t+1} = x_i^t + \alpha_t\Big[ - \tilde{G}_i(z(t), z(t+1))x_i^t + h_i(z(t))\Big]. \tag{17}$$

Let $P^\theta$ be the transition matrix and the state-action pair when the policy is $\theta$. Because at time $t$, the policy is $\theta(t)$, as such the transition matrix from $z(t)$ to $z(t+1)$ is $P^{\theta(t)}$. We define

$$G_i^\theta(z) = \mathbb{E}_{z'\sim P^\theta(\cdot|z)}\tilde{G}_i(z, z') = \begin{bmatrix} 1 & 0 \\ \frac{1}{c'}\mathbf{e}_{z_{N_i^\kappa}} & \mathbf{e}_{z_{N_i^\kappa}}[\mathbf{e}_{z_{N_i^\kappa}}^\top - P^\theta(\cdot|z)\Phi_i] \end{bmatrix} \tag{18}$$

where $P^\theta(\cdot|z)$ is understood as the $z$'th row of $P^\theta$ and is treated as a row vector. Also, we have defined $\Phi_i \in \mathbb{R}^{\mathcal{Z}\times\hat{\mathcal{Z}}_{N_i^\kappa}}$ to be a matrix with each row indexed by $z \in \mathcal{Z}$ and each column indexed by $z'_{N_i^\kappa} \in \hat{\mathcal{Z}}_{N_i^\kappa}$, and its entries are given by $\Phi_i(z, z'_{N_i^\kappa}) = 1$ if $z_{N_i^\kappa} = z'_{N_i^\kappa}$ and $\Phi_i(z, z'_{N_i^\kappa}) = 0$ elsewhere. Then, (17) can be rewritten as,

$$x_i^{t+1} = x_i^t + \alpha_t\Big[ - G_i^{\theta(t)}(z(t))x_i^t + h_i(z(t)) + \underbrace{[G_i^{\theta(t)}(z(t)) - \tilde{G}_i(z(t), z(t+1))]}_{:=\xi_i^{t+1}} x_i^t\Big]. \tag{19}$$

This will correspond to the first equation in the stochastic approximation scheme (14) that we reviewed in Appendix C.1. Further, the actor update can be written as,

$$\theta(t+1) = \theta(t) + \eta_t \Gamma(\hat{\mathbf{Q}}^{\mathbf{t}})\hat{g}(t). \tag{20}$$

with $\hat{g}_i(t) = \nabla_{\theta_i} \log \zeta_i^{\theta_i(t)}(a_i(t)|s_i(t))\frac{1}{n}\sum_{j \in N_i^\kappa} \hat{Q}_j^t(z_{N_j}(t))$. We identify equation (19) and (20) with the stochastic approximation scheme in (14), where $x_i^t, G_i^\theta, h_i, \xi_i^{t+1}, \Gamma(\hat{\mathbf{Q}}^{\mathbf{t}})\hat{g}(t)$ are identified with the $x^t, G^\theta, h^\theta, \xi^{t+1}, H^{t+1}$ in (14) respectively. In what follows, we will check all the assumptions (Assumption 6 to 11) in Appendix C.1 and invoke Theorem 4.

To that end, we first define $\bar{G}_i(\theta), \bar{h}_i(\theta), \hat{G}_i^\theta(z), \hat{h}_i^\theta(z)$, which will be the solution to the Poisson equation in Assumption 8(a). Given $\theta$, recall the stationary distribution under policy $\theta$ is $\pi^\theta$ and matrix $D^\theta = \text{diag}(\pi^\theta)$. We define,

$$\bar{G}_i(\theta) = \mathbb{E}_{z\sim\pi^\theta} G_i^\theta(z) = \begin{bmatrix} 1 & 0 \\ \frac{1}{c'}\Phi_i^\top \pi^\theta & \Phi_i^\top D^\theta[\Phi_i - P^\theta \Phi_i] \end{bmatrix},$$

$$\bar{h}_i(\theta) = \mathbb{E}_{z\sim\pi^\theta} h_i(z) = \begin{bmatrix} c'(\pi^\theta)^\top r_i \\ \Phi_i^\top D^\theta r_i \end{bmatrix},$$

where in the last line, $r_i$ is understood as a vector over the entire state-action space $\mathcal{Z}$, (though it only depends on $z_i$). We also define,

$$\hat{G}_i^\theta(z) = \mathbb{E}_\theta[\sum_{t=0}^\infty [G_i^\theta(z(t)) - \bar{G}_i(\theta)]|z(0) = z],$$

$$\hat{h}_i^\theta(z) = \mathbb{E}_\theta[\sum_{t=0}^\infty [h_i(z(t)) - \bar{h}_i(\theta)]|z(0) = z].$$

It is easy to check that the above definitions will be the solution to the Poisson equation in Assumption 8(a).

We will now start to check all the assumptions. We will frequently use the following auxiliary lemma, which is an immediate consequence of Assumption 4.

**Lemma 6.** *Under Assumption 4, for vector $d \in \mathbb{R}^{\mathcal{Z}}$ such that $\mathbf{1}^\top d = 0$, we have, $\|((P^\theta)^\top)^t d\|_1 \le c_\infty \mu_D^t \|d\|_1$ for $c_\infty = \sqrt{\frac{|\mathcal{Z}|}{\sigma}}$.*

*Proof.* As $P^\theta$ is a ergodic stochastic matrix with stationary distribution $\pi^\theta$, we have $(P^\theta - \mathbf{1}(\pi^\theta)^\top)^t = (P^\theta)^t - \mathbf{1}(\pi^\theta)^\top$. As a result,

$$((P^\theta)^\top)^t d = [((P^\theta)^\top)^t - \pi^\theta \mathbf{1}^\top]d = [(P^\theta)^t - \mathbf{1}(\pi^\theta)^\top]^\top d = [(P^\theta - \mathbf{1}(\pi^\theta)^\top)^t]^\top d.$$

As a result, by Assumption 4, $\|((P^\theta)^\top)^t d\|_{D^\theta} \le \|[P^\theta - \mathbf{1}(\pi^\theta)^\top]^\top\|_{D^\theta}^t \|d\|_{D^\theta} \le \mu_D^t \|d\|_{D^\theta}$. The rest follows from a change of norm as $\sqrt{\frac{\sigma}{|\mathcal{Z}|}}\|d\|_1 \le \sqrt{\sigma}\|d\|_2 \le \|d\|_{D^\theta} \le \|d\|_2 \le \|d\|_1$. $\qquad\square$

**Checking Assumptions 6, 7 and 8.** Clearly Assumption 6, Assumption 7 and Assumption 8(a) are satisfied. To check Assumption 8(b) and (c), we have the following Lemma.

**Lemma 7.** *(a) For any $z, z' \in \mathcal{Z}$, we have,*

$$\|\tilde{G}_i(z, z')\|_\infty \le 2 + \frac{1}{c'} := G_{\max}, \quad \|h_i(z)\|_\infty \le \max(c', 1)\bar{r} := h_{\max}.$$

*As a result, $\|G_i^\theta(z)\|_\infty \le G_{\max}$ and $\|\bar{G}_i(\theta)\|_\infty \le G_{\max}, \|\bar{h}_i(\theta)\|_\infty \le h_{\max}$.*

*(b) We also have that,*

$$\left\| \mathbb{E}_\theta [G_i^\theta(z(t)) - \bar{G}_i(\theta)|z(0) = z] \right\|_\infty \le 2G_{\max}c_\infty \mu_D^t,$$

$$\left\| \mathbb{E}_\theta [h_i(z(t)) - \bar{h}_i(\theta)|z(0) = z] \right\|_\infty \le 2h_{\max}c_\infty \mu_D^t.$$

*As a consequence, for any $z$, $\|\hat{G}_i^\theta(z)\|_\infty \le 2G_{\max}c_\infty \frac{1}{1-\mu_D}, \|\hat{h}_i^\theta(z)\|_\infty \le 2h_{\max}c_\infty \frac{1}{1-\mu_D}$.*

*Proof.* Part (a) follows directly from the definition as well as the bounded reward (Assumption 1). Part (b) is a consequence of Lemma 6. In details, given $z$, let $d^t$ be the distribution of $z(t)$ starting form $z$. Then,

$$
\begin{aligned}
\left\| \mathbb{E}_\theta \left[ G_i^\theta(z(t)) - \bar{G}_i(\theta) | z(0) = z \right] \right\|_\infty &= \| \mathbb{E}_{z \sim d^t} G_i^\theta(z) - \mathbb{E}_{z \sim \pi^\theta} G_i^\theta(z) \|_\infty \\
&= \| \sum_z (d^t(z) - \pi^\theta(z)) G_i^\theta(z) \|_\infty \\
&\leq \sum_z |d^t(z) - \pi^\theta(z)| \| G_i^\theta(z) \|_\infty \\
&\leq G_{\max} \| d^t - \pi^\theta \|_1 \\
&= G_{\max} \| ((P^\theta)^\top)^t (d^0 - \pi^\theta) \|_1 \leq G_{\max} 2 c_\infty \mu_D^t.
\end{aligned}
$$

The proof for $h_i$ is similar. $\qquad\square$

Next, the following Lemma 8 shows the Lipschitz condition in Assumption 8 (d) and (e) are true. The proof of Lemma 8 is postponed to Appendix C.3

**Lemma 8.** *The following holds.*

(a) $P^\theta$ *and* $\pi^\theta$ *are Lipschitz in* $\theta$.

(b) $\bar{G}_i(\theta)$ *and* $\bar{h}_i(\theta)$ *are Lipschitz in* $\theta$.

(c) *For any* $z$, $[P^\theta \hat{h}_i^\theta](z)$ *and* $[P^\theta \hat{G}_i^\theta](z)$ *are Lipschitz in* $\theta$ *with the Lipschitz constant independent of* $z$.

**Checking Assumption 9.** Recall that $\theta(t+1) = \theta(t) + \eta_t \Gamma(\hat{\mathbf{Q}}^t) \hat{g}(t)$. Note that $\| \hat{g}_i(t) \| \leq L_i \max_j \| \hat{Q}_j^t \|_\infty$. By the definition of $\Gamma(\hat{\mathbf{Q}}^t)$, we have almost surely $\| \Gamma(\hat{\mathbf{Q}}^t) \hat{g}_i(t) \| \leq L_i$ for all $t$. As such, almost surely, for all $t$, $\| \Gamma(\hat{\mathbf{Q}}^t) \hat{g}(t) \| \leq L$. This, together with our selection of $\eta_t$ (Assumption 3), shows that Assumption 9 is satisfied.

**Checking Assumption 10.** Recall that $\xi_i^{t+1} = G_i^{\theta(t)}(z(t)) - \tilde{G}_i(z(t), z(t+1))$. We have clearly $\mathbb{E}\xi_i^{t+1} | \mathcal{F}_t = 0$ per the definition of $G_i^\theta(z)$. Further, $\| \xi_i^{t+1} \|_\infty \leq 2 G_{\max}$. So Assumption 10 is satisfied.

**Checking Assumption 11.** Finally, we check Assumption 11, the assumption that $\bar{G}_i(\theta)$ is uniformly positive definite. This is done in the following Lemma 9, whose proof is postponed to Appendix C.4.

**Lemma 9.** *We have when* $c' = \frac{1}{\sigma \sqrt{(1 - \mu_D)}}$, *then for any* $\theta$, $x_i^\top \bar{G}_i(\theta) x_i \geq \frac{1}{2}(1 - \mu_D)\sigma^2 \| x_i \|^2$.

Given $\theta$, let $x_i^\theta = [c' \hat{\mu}_i^\theta; \hat{Q}_i^\theta]$ be the unique solution to $\bar{h}_i(\theta) - \bar{G}_i(\theta) x_i = 0$. Now that Assumptions 6 to 11 are satisfied, by Theorem 4 we immediately have almost surely $\lim_{t \to \infty} \| x_i^t - [\bar{G}_i(\theta(t))]^{-1} \bar{h}_i(\theta(t)) \| = 0$, and $\sup_{t \geq 0} \| x_i^t \| < \infty$. As $x_i^t = [c' \hat{\mu}_i^t; \hat{Q}_i^t]$, this directly implies $\lim_{t \to \infty} \hat{Q}_i^t - \hat{Q}_i^{\theta(t)} = 0$ and $\sup_{t \geq 0} \| \hat{Q}_i^t \|_\infty \leq \infty$. This proves part (b) of Theorem 3. For part (a), we show the following Lemma 10 on the property of $x_i^\theta$, whose proof is postponed to Appendix C.5. With Lemma 10, the proof of Theorem 3 is concluded.

**Lemma 10.** *Given* $\theta$, *the solution* $x_i^\theta = [c' \hat{\mu}_i^\theta; \hat{Q}_i^\theta]$ *to* $\bar{h}_i(\theta) - \bar{G}_i(\theta) x_i = 0$ *satisfies* $\hat{\mu}_i^\theta = J_i(\theta)$. *Further, there exists some* $c_i^\theta \in \mathbb{R}$ *s.t.*

$$
\| \Phi_i \hat{Q}_i^\theta + c_i^\theta \mathbf{1} - Q_i^\theta \|_{D^\theta} \leq \frac{c \rho^{\kappa + 1}}{1 - \mu_D}, \tag{21}
$$

*where* $\mathbf{1}$ *is the all one vector in* $\mathbb{R}^{\mathcal{Z}}$.

### C.3 Proof of Lemma 8

To show (a), notice that, $P^\theta(s', a' | s, a) = P(s' | s, a) \zeta^\theta(a' | s')$. Therefore,

$$
\| P^\theta - P^{\bar{\theta}} \|_\infty = \max_{s,a} \sum_{s',a'} P(s' | s, a) |\zeta^\theta(a' | s') - \zeta^{\bar{\theta}}(a' | s')|
$$

$$\leq L\|\theta - \bar{\theta}\| \max_{s,a} \sum_{s',a'} P(s'|s,a)$$

$$= L|\mathcal{A}|\|\theta - \bar{\theta}\| := L_P\|\theta - \bar{\theta}\|,$$

where in the inequality, we have used that for any $a \in \mathcal{A}$, $s \in \mathcal{S}$, as $\|\nabla_{\theta_i} \log \zeta^\theta(a|s)\| = \|\nabla_{\theta_i} \log \zeta_i^{\theta_i}(a_i|s_i)\| \leq L_i$ (Assumption 5), we have $\|\nabla_\theta \zeta^\theta(a|s)\| \leq \|\nabla_\theta \log \zeta^\theta(a|s)\| \leq \sqrt{\sum_{i\in\mathcal{N}} L_i^2} = L$.

Next, we show $\pi^\theta$ is Lipschitz continuous in $\theta$. Notice that $\pi^\theta$ satisfies $\pi^\theta = (P^\theta)^\top \pi^\theta$. As such, we have,

$$\pi^\theta - \pi^{\bar{\theta}} = (P^\theta)^\top(\pi^\theta - \pi^{\bar{\theta}}) + (P^\theta - P^{\bar{\theta}})^\top \pi^{\bar{\theta}}$$

$$= ((P^\theta)^\top)^k(\pi^\theta - \pi^{\bar{\theta}}) + \sum_{\ell=0}^{k-1}((P^\theta)^\top)^\ell(P^\theta - P^{\bar{\theta}})^\top \pi^{\bar{\theta}} = \sum_{\ell=0}^{\infty}((P^\theta)^\top)^\ell(P^\theta - P^{\bar{\theta}})^\top \pi^{\bar{\theta}}.$$

Notice that by Lemma 6,

$$\|((P^\theta)^\top)^\ell(P^\theta - P^{\bar{\theta}})^\top \pi^{\bar{\theta}}\|_1 \leq c_\infty \mu_D^\ell \|(P^\theta - P^{\bar{\theta}})^\top \pi^{\bar{\theta}}\|_1 \leq c_\infty \mu_D^\ell \|P^\theta - P^{\bar{\theta}}\|_\infty.$$

Therefore, we have

$$\|\pi^\theta - \pi^{\bar{\theta}}\|_1 \leq \frac{c_\infty}{1-\mu_D}\|P^\theta - P^{\bar{\theta}}\|_\infty \leq \frac{c_\infty}{1-\mu_D}L_P\|\theta - \bar{\theta}\|.$$

So we are done for part (a).

For part (b), notice that $\bar{h}_i(\theta)$ depends on $\theta$ only through $\pi^\theta$ and is linear in $\pi^\theta$. As a result $\bar{h}_i(\theta)$ is Lipschitz in $\theta$. For similar reasons, for $\bar{G}_i(\theta)$ we only need to show $D^\theta P^\theta$ is Lipschitz in $\theta$. This is true because both $D^\theta$ and $P^\theta$ are Lipschitz in $\theta$, and they themselves are bounded.

For part (c), fixing any initial $z$, let $d^{\theta,t}$ be the distribution of $z(t)$ under policy $\theta$. We first show that $d^{\theta,t} - \pi^\theta$ is Lipschitz in $\theta$ with Lipschitz constant geometrically decaying in $t$. To this end, note that

$$d^{\theta,t} - \pi^\theta - (d^{\bar{\theta},t} - \pi^{\bar{\theta}}) = (P^\theta)^\top(d^{\theta,t-1} - \pi^\theta) - (P^{\bar{\theta}})^\top(d^{\bar{\theta},t-1} - \pi^{\bar{\theta}})$$

$$= (P^\theta)^\top[d^{\theta,t-1} - \pi^\theta - (d^{\bar{\theta},t-1} - \pi^{\bar{\theta}})] + (P^\theta - P^{\bar{\theta}})^\top(d^{\bar{\theta},t-1} - \pi^{\bar{\theta}})$$

$$= ((P^\theta)^t)^\top(\pi^{\bar{\theta}} - \pi^\theta) + \sum_{\ell=0}^{t-1}((P^\theta)^\ell)^\top(P^\theta - P^{\bar{\theta}})^\top(d^{\bar{\theta},t-\ell-1} - \pi^{\bar{\theta}}).$$

As such, we have,

$$\|d^{\theta,t} - \pi^\theta - (d^{\bar{\theta},t} - \pi^{\bar{\theta}})\|_1 \leq c_\infty \mu_D^t\|\pi^{\bar{\theta}} - \pi^\theta\|_1 + \sum_{\ell=0}^{t-1} c_\infty \mu_D^\ell \|P^\theta - P^{\bar{\theta}}\|_\infty \|d^{\bar{\theta},t-\ell-1} - \pi^{\bar{\theta}}\|_1$$

$$\leq c_\infty \mu_D^t \frac{c_\infty}{1-\mu_D}L_P\|\theta - \bar{\theta}\| + \sum_{\ell=0}^{t-1} c_\infty \mu_D^\ell L_P\|\theta - \bar{\theta}\| 2c_\infty \mu_D^{t-\ell-1}$$

$$= \frac{c_\infty^2 L_P}{1-\mu_D}\mu_D^t\|\theta - \bar{\theta}\| + 2c_\infty^2 L_P t \mu_D^{t-1}\|\theta - \bar{\theta}\|$$

$$< \frac{5c_\infty^2 L_P}{1-\mu_D}\left(\frac{1+\mu_D}{2}\right)^t\|\theta - \bar{\theta}\|. \tag{22}$$

Next, we turn to $\hat{G}_i^\theta(z)$ and show its Lipschitz continuity in $\theta$. Note that by definition,

$$\hat{G}_i^\theta(z) = \mathbb{E}_\theta\left[\sum_{t=0}^{\infty}[G_i^\theta(z(t)) - \bar{G}_i(\theta)]\Big|z(0) = z\right] = \sum_{t=0}^{\infty}[\mathbb{E}_{z'\sim d^{\theta,t}}G_i^\theta(z') - \mathbb{E}_{z'\sim \pi^\theta}G_i^\theta(z')]$$

$$= \sum_{t=0}^{\infty}\sum_{z'\in\mathcal{Z}}(d^{\theta,t}(z') - \pi^\theta(z'))G_i^\theta(z').$$

As such,

$$\|\hat{G}_i^\theta(z) - \hat{G}_i^{\bar\theta}(z)\|_\infty$$

$$\leq \sum_{t=0}^\infty \sum_{z' \in \mathcal{Z}} \left\| (d^{\theta,t}(z') - \pi^\theta(z'))G_i^\theta(z') - (d^{\bar\theta,t}(z') - \pi^{\bar\theta}(z'))G_i^{\bar\theta}(z') \right\|_\infty$$

$$\leq \sum_{t=0}^\infty \sum_{z' \in \mathcal{Z}} \left[ |d^{\theta,t}(z') - \pi^\theta(z') - (d^{\bar\theta,t}(z') - \pi^{\bar\theta}(z'))| \|G_i^\theta(z')\|_\infty + |d^{\bar\theta,t}(z') - \pi^{\bar\theta}(z')| \|G_i^\theta(z') - G_i^{\bar\theta}(z')\|_\infty \right]$$

$$\leq \sum_{t=0}^\infty \left[ \|d^{\theta,t} - \pi^\theta - (d^{\bar\theta,t} - \pi^{\bar\theta})\|_1 G_{\max} + \|d^{\bar\theta,t} - \pi^{\bar\theta}\|_1 \sup_{z'} \|G_i^\theta(z') - G_i^{\bar\theta}(z')\|_\infty \right]$$

$$\leq \sum_{t=0}^\infty \left[ \frac{5c_\infty^2 L_P G_{\max}}{1 - \mu_D} (\frac{1 + \mu_D}{2})^t \|\theta - \bar\theta\| + 2c_\infty \mu_D^t \sup_{z'} \|G_i^\theta(z') - G_i^{\bar\theta}(z')\|_\infty \right].$$

Since $G_i^\theta(z')$ depends on $\theta$ only through $P^\theta$ and is linear in $P^\theta$, $G_i^\theta(z')$ is Lipschitz in $\theta$. Therfore, in the above summation, each summand can be written as some geometrically decaying term times $\|\theta - \bar\theta\|$. As such, $\hat{G}_i^\theta(z)$ is Lipschitz in $\theta$, and the Lipschitz constant can be made independent of $z$ by taking the sup over the finite set $z \in \mathcal{Z}$. As a result, $[P^\theta \hat{G}_i^\theta](z) = \sum_{z'} P^\theta(z'|z)\hat{G}_i^\theta(z')$ is Lipschitz in $\theta$ as well since both $P^\theta$ and $\hat{G}_i^\theta(z')$ are Lipschitz in $\theta$ and bounded.

The proof for the Lipschitz continuity of $P^\theta \hat{h}_i^\theta(z)$ is similar and is hence omitted. Therefore, part (c) is done and the proof is concluded.

### C.4 Proof of Lemma 9

Recall that,

$$\bar{G}_i(\theta) = \left[ \begin{array}{cc} 1 & 0 \\ \frac{1}{c'} \Phi_i^\top \pi^\theta & \Phi_i^\top D^\theta \left[ \Phi_i - P^\theta \Phi_i \right] \end{array} \right].$$

Let $x_i = [\hat{\mu}_i, \hat{Q}_i]$ and define $\hat{\Phi}_i = \Phi_i - \mathbf{1}(\pi^\theta)^\top \Phi_i$. Then,

$$\Phi_i^\top D^\theta \Phi_i = \hat{\Phi}_i^\top D^\theta \hat{\Phi}_i + \Phi_i^\top \pi^\theta \mathbf{1}^\top D^\theta \hat{\Phi}_i + \hat{\Phi}_i^\top D^\theta \mathbf{1}(\pi^\theta)^\top \Phi_i + \Phi_i^\top \pi^\theta \mathbf{1}^\top D^\theta \mathbf{1}(\pi^\theta)^\top \Phi_i$$

$$= \hat{\Phi}_i^\top D^\theta \hat{\Phi}_i + \Phi_i^\top \pi^\theta (\pi^\theta)^\top \Phi_i,$$

$$\Phi_i^\top D^\theta P^\theta \Phi_i = \hat{\Phi}_i^\top D^\theta P^\theta \hat{\Phi}_i + \Phi_i^\top \pi^\theta \mathbf{1}^\top D^\theta P^\theta \hat{\Phi}_i + \hat{\Phi}_i^\top D^\theta P^\theta \mathbf{1}(\pi^\theta)^\top \Phi_i + \Phi_i^\top \pi^\theta \mathbf{1}^\top D^\theta P^\theta \mathbf{1}(\pi^\theta)^\top \Phi_i$$

$$= \hat{\Phi}_i^\top D^\theta P^\theta \hat{\Phi}_i + \Phi_i^\top \pi^\theta (\pi^\theta)^\top \Phi_i.$$

As such,

$$\Phi_i^\top D^\theta \Phi_i - \Phi_i^\top D^\theta P^\theta \Phi_i = \hat{\Phi}_i^\top D^\theta \hat{\Phi}_i - \hat{\Phi}_i^\top D^\theta P^\theta \hat{\Phi}_i,$$

from which, we have using Assumption 4,

$$\hat{Q}_i^\top (\Phi_i^\top D^\theta \Phi_i - \Phi_i^\top D^\theta P^\theta \Phi_i)\hat{Q}_i \geq \|\hat{\Phi}_i \hat{Q}_i\|_{D^\theta}^2 - \|\hat{\Phi}_i \hat{Q}_i\|_{D^\theta} \|P^\theta \hat{\Phi}_i \hat{Q}_i\|_{D^\theta}$$

$$= \|\hat{\Phi}_i \hat{Q}_i\|_{D^\theta}^2 - \|\hat{\Phi}_i \hat{Q}_i\|_{D^\theta} \|(P^\theta - \mathbf{1}(\pi^\theta)^\top)\hat{\Phi}_i \hat{Q}_i\|_{D^\theta}$$

$$\geq \|\hat{\Phi}_i \hat{Q}_i\|_{D^\theta}^2 - \mu_D \|\hat{\Phi}_i \hat{Q}_i\|_{D^\theta}^2$$

$$= (1 - \mu_D)\hat{Q}_i^\top \hat{\Phi}_i^\top D^\theta \hat{\Phi}_i \hat{Q}_i$$

$$= (1 - \mu_D)\hat{Q}_i^\top (\Phi_i^\top D^\theta \Phi_i - \Phi_i^\top \pi^\theta (\pi^\theta)^\top \Phi_i)\hat{Q}_i$$

$$\geq (1 - \mu_D)\sigma^2 \|\hat{Q}_i\|^2, \tag{23}$$

where the last step is due to the following. Let $v \in \mathbb{R}^{\hat{\mathcal{Z}}_{N_i^\kappa}}$ be the marginalized distribution of $z_{N_i^\kappa} \in \hat{\mathcal{Z}}_{N_i^\kappa}$ under $\pi^\theta$, i.e. $v(z_{N_i^\kappa}) = \pi^\theta(z_{N_i^\kappa})$. Using $v(z_{N_i^\kappa}) \geq \sigma$ and $\sum_{z_{N_i^\kappa} \in \hat{\mathcal{Z}}_{N_i^\kappa}} v(z_{N_i^\kappa}) \leq 1 - \sigma$ (Assumption 4), we have,

$$\Phi_i^\top D^\theta \Phi_i - \Phi_i^\top \pi^\theta (\pi^\theta)^\top \Phi_i = \text{diag}(v) - vv^\top = \text{diag}(v)^{\frac{1}{2}}(I - \text{diag}(v)^{-\frac{1}{2}} v(\text{diag}(v)^{-\frac{1}{2}} v)^\top)\text{diag}(v)^{\frac{1}{2}}$$

$$\succeq (1 - \|\mathrm{diag}(v)^{-\frac{1}{2}}v\|^2)\mathrm{diag}(v)$$
$$\succeq \sigma^2 I.$$

Building on (23), the rest of the proof follows easily. We have,

$$x_i^\top \bar{G}_i(\theta) x_i \geq \hat{\mu}_i^2 + (1 - \mu_D)\sigma^2 \|\hat{Q}_i\|^2 + \frac{1}{c'}\hat{Q}_i^\top \Phi_i^\top \pi^\theta \hat{\mu}_i$$

$$\geq \hat{\mu}_i^2 + (1 - \mu_D)\sigma^2 \|\hat{Q}_i\|^2 - \frac{1}{c'}\|\hat{Q}_i\||\hat{\mu}_i|$$

$$\geq \min(\frac{1}{2}, \frac{1}{2}(1 - \mu_D)\sigma^2)\|x_i\|^2,$$

where we have used

$$\frac{1}{2}\hat{\mu}_i^2 + \frac{1}{2}(1 - \mu_D)\sigma^2 \|\hat{Q}_i\|^2 \geq \sigma\sqrt{(1 - \mu_D)}\|\hat{Q}_i\||\hat{\mu}_i| \geq \frac{1}{c'}\|\hat{Q}_i\||\hat{\mu}_i|.$$

## C.5 Proof of Lemma 10

By the definition of $\bar{G}_i(\theta)$ and $\bar{h}_i(\theta)$, we have $\hat{\mu}_i^\theta = (\pi^\theta)^\top r_i = J_i(\theta)$, the average reward at node $i$ under policy $\theta$, and $\hat{Q}_i^\theta \in \mathbb{R}^{\hat{\mathcal{Z}}_{N_i^\kappa}}$ is the solution to the following linear equation (the solution must be unique due to Lemma 9),

$$0 = -\Phi_i^\top D^\theta \hat{\mu}_i^\theta \mathbf{1} + \Phi_i^\top D^\theta [P^\theta \Phi_i - \Phi_i]\hat{Q}_i^\theta + \Phi_i^\top D^\theta r_i$$
$$= \Phi_i^\top D^\theta [r_i - \hat{\mu}_i^\theta \mathbf{1} + P^\theta \Phi_i \hat{Q}_i^\theta] - \Phi_i^\top D^\theta \Phi_i \hat{Q}_i^\theta$$
$$= \Phi_i^\top D^\theta [r_i - J_i(\theta)\mathbf{1} + P^\theta \Phi_i \hat{Q}_i^\theta] - \Phi_i^\top D^\theta \Phi_i \hat{Q}_i^\theta. \tag{24}$$

To understand the solution of (24), we define an equivalent expanded equation, whose solution can be related to the Bellman operator. For this purpose, define $\tilde{\Phi}_i \in \mathbb{R}^{\mathcal{Z} \times \mathcal{Z}_{N_i^\kappa}}$ to be a matrix with each row indexed by $z \in \mathcal{Z}$ and each column indexed by $z'_{N_i^\kappa} \in \mathcal{Z}_{N_i^\kappa}$ and $\tilde{\Phi}_i(z, z'_{N_i^\kappa}) = 1$ if $z_{N_i^\kappa} = z'_{N_i^\kappa}$ and 0 elsewhere. In other words, $\tilde{\Phi}_i$ is essentially $\Phi_i$ with the additional column corresponding to the dummy state-action pair $\tilde{z}_{N_i^\kappa}$. Consider the following equations on $\bar{Q}_i^\theta \in \mathbb{R}^{\mathcal{Z}_{N_i^\kappa}}$

$$0 = \tilde{\Phi}_i^\top D^\theta [r_i - J_i(\theta)\mathbf{1} + P^\theta \tilde{\Phi}_i \bar{Q}_i^\theta] - \tilde{\Phi}_i^\top D^\theta \tilde{\Phi}_i \bar{Q}_i^\theta, \tag{25a}$$
$$0 = \bar{Q}_i^\theta(\tilde{z}_{N_i^\kappa}). \tag{25b}$$

**Claim 1:** The equations (24) and (25) are equivalent in the sense that both have unique solutions, and the solutions are related by $\hat{Q}_i^\theta(z_{N_i^\kappa}) = \bar{Q}_i^\theta(z_{N_i^\kappa}), \forall z_{N_i^\kappa} \in \hat{\mathcal{Z}}_{N_i^\kappa}$.

Before we prove the claim, we first show (25a) can be actually reformulated as the fixed point equation related to the Bellman operator.

**Reformulation of** (25a) **as fixed point equation**. It is easy to check that $\tilde{D}_i^\theta = \tilde{\Phi}_i^\top D^\theta \tilde{\Phi}_i \in \mathbb{R}^{\mathcal{Z}_{N_i^\kappa} \times \mathcal{Z}_{N_i^\kappa}}$ is a diagonal matrix, and the $z_{N_i^\kappa}$'th diagonal entry is the marginal probability of $z_{N_i^\kappa}$ under $\pi^\theta$, which is non-zero by Assumption 4. Therefore, $\tilde{\Phi}_i^\top D^\theta \tilde{\Phi}_i$ is invertable and matrix $\Pi_i^\theta = (\tilde{\Phi}_i^\top D^\theta \tilde{\Phi}_i)^{-1}\tilde{\Phi}_i^\top D^\theta$ is well defined. Further, the $z_{N_i^\kappa}$'th row of $\Pi_i^\theta$ is in fact the conditional distribution of the full state $z$ given $z_{N_i^\kappa}$. So, $\Pi_i^\theta$ must be a stochastic matrix and is non-expansive in infinity norm. Let $\mathrm{TD}_i^\theta(Q_i) = r_i - J_i(\theta)\mathbf{1} + P^\theta Q_i$ be the Bellman operator for reward $r_i$. Further, define operator $g : \mathbb{R}^{\mathcal{Z}_{N_i^\kappa}} \to \mathbb{R}^{\mathcal{Z}_{N_i^\kappa}}$ given by $g(\tilde{Q}_i) = \Pi_i^\theta \mathrm{TD}_i^\theta \tilde{\Phi}_i \tilde{Q}_i$ for $\tilde{Q}_i \in \mathbb{R}^{\mathcal{Z}_{N_i^\kappa}}$. Then, (25a) is equvalent to the fixed point equation of operator $g$, $g(\bar{Q}_i^\theta) = \bar{Q}_i^\theta$. Our next claim studies the structure of the fixed points of $g$.

**Claim 2:** Define $\Xi^\theta = \{Q_i \in \mathbb{R}^{\mathcal{Z}} : \mathbb{E}_{z \sim \pi^\theta} Q_i(z) = 0\}$, and $\tilde{\Xi}_i^\theta = \{\tilde{Q}_i \in \mathbb{R}^{\mathcal{Z}_{N_i^\kappa}} : \mathbb{E}_{z \sim \pi^\theta} \tilde{Q}_i(z_{N_i^\kappa}) = 0\}$. We claim that $g$ has a unique fixed point within $\tilde{\Xi}_i^\theta$ which we denote as $\tilde{Q}_i^\theta$. Further, all fixed points of $g$ are the set $\{\tilde{Q}_i^\theta + c_i \mathbf{1} : c_i \in \mathbb{R}\}$.

**Proof of Claim 2.** We in fact show $g$ maps $\tilde{\Xi}_i^\theta$ to $\tilde{\Xi}_i^\theta$ and is a contraction in $\| \cdot \|_{\tilde{D}_i^\theta}$ norm when restricted to $\tilde{\Xi}_i^\theta$, which will guarantee the existence and uniqueness of $\tilde{Q}_i^\theta$. To see this, we check the following steps.

- $\tilde{\Phi}_i$ maps $\tilde{\Xi}_i^\theta$ to $\Xi^\theta$ and preserves metric from $\|\cdot\|_{\tilde{D}_i^\theta}$ to $\|\cdot\|_{D^\theta}$. To see this, note that $\tilde{\Phi}_i^\top D^\theta \tilde{\Phi}_i = \tilde{D}_i^\theta$.

- $\mathrm{TD}_i^\theta$ maps $\Xi^\theta$ to $\Xi^\theta$ and further, it is a $\mu_D$ contraction in $\|\cdot\|_{D^\theta}$ when restricted to $\Xi^\theta$. To see this, note that for $Q_i, Q_i' \in \Xi^\theta$, $\|\mathrm{TD}_i^\theta(Q_i) - \mathrm{TD}_i^\theta(Q_i')\|_{D^\theta} = \|P^\theta(Q_i - Q_i')\|_{D^\theta} = \|(P^\theta - \mathbf{1}(\pi^\theta)^\top)(Q_i - Q_i')\|_{D^\theta} \le \mu_D \|Q_i - Q_i'\|_{D^\theta}$, where we have used Assumption 4.

- $\Pi_i^\theta$ maps $\Xi^\theta$ to $\tilde{\Xi}_i^\theta$ and is non-expensive from $\|\cdot\|_{D^\theta}$ to $\|\cdot\|_{\tilde{D}_i^\theta}$. To see this, notice that $(\Pi_i^\theta Q_i)(z_{N_i^\kappa}) = \mathbb{E}_{z' \sim \pi^\theta(z'|z'_{N_i^\kappa}=z_{N_i^\kappa})} Q_i(z')$. As such, when $Q_i \in \Xi^\theta$, $\mathbb{E}_{z \sim \pi^\theta}(\Pi_i^\theta Q_i)(z_{N_i^\kappa}) = \mathbb{E}_{z' \sim \pi^\theta} Q_i(z') = 0$, which shows $\Pi_i^\theta Q_i \in \tilde{\Xi}_i^\theta$. Finally, one can check $\Pi_i^\theta$ is non-expensive from $\|\cdot\|_{D^\theta}$ to $\|\cdot\|_{\tilde{D}_i^\theta}$ by noting $(\tilde{D}_i^\theta)^{1/2} \Pi_i^\theta (D^\theta)^{-1/2} = (\tilde{D}_i^\theta)^{-1/2} \tilde{\Phi}_i^\top (D^\theta)^{1/2}$, the rows of which are orthornormal vectors.

Combining these relations, $g$ maps $\tilde{\Xi}_i^\theta$ to itself and further, we have for $\tilde{Q}_i, \tilde{Q}_i' \in \tilde{\Xi}_i^\theta$,

$$\|g(\tilde{Q}_i) - g(\tilde{Q}_i')\|_{\tilde{D}_i^\theta} = \|\Pi_i^\theta(\mathrm{TD}_i^\theta \tilde{\Phi}_i \tilde{Q}_i - \mathrm{TD}_i^\theta \tilde{\Phi}_i \tilde{Q}_i')\|_{\tilde{D}_i^\theta} \le \|\mathrm{TD}_i^\theta(\tilde{\Phi}_i \tilde{Q}_i) - \mathrm{TD}_i^\theta(\tilde{\Phi}_i \tilde{Q}_i')\|_{D^\theta}$$
$$\le \mu_D \|\tilde{\Phi}_i(\tilde{Q}_i - \tilde{Q}_i')\|_{D^\theta} = \mu_D \|\tilde{Q}_i - \tilde{Q}_i'\|_{\tilde{D}_i^\theta},$$

which shows $g$ is a contraction when restricted to $\tilde{\Xi}_i^\theta$. This shows $g$ has a unique fixed point within $\tilde{\Xi}_i^\theta$, which we denote by $\tilde{Q}_i^\theta$. Further, note for any $c_i \in \mathbb{R}$,

$$g(\tilde{Q}_i + c_i \mathbf{1}) = \Pi_i^\theta \mathrm{TD}_i^\theta \tilde{\Phi}_i(\tilde{Q}_i + c_i \mathbf{1}) = \Pi_i^\theta \mathrm{TD}_i^\theta(\tilde{\Phi}_i \tilde{Q}_i + c_i \mathbf{1})$$
$$= \Pi_i^\theta[\mathrm{TD}_i^\theta \tilde{\Phi}_i \tilde{Q}_i + c_i \mathbf{1}] = g(\tilde{Q}_i) + c_i \mathbf{1}.$$

Therefore, let $\tilde{Q}_i$ be a fixed point of $g$, then $\tilde{Q}_i - \mathbf{1}\mathbb{E}_{z \sim \pi^\theta} \tilde{Q}_i(z_{N_i^\kappa})$ will be a fixed point of $g$ within $\tilde{\Xi}_i^\theta$. As such, the set of fixed point of $g$ can be written in the form $\{\tilde{Q}_i^\theta + c_i \mathbf{1} : c_i \in \mathbb{R}\}$. $\square$

We are now ready to prove Claim 1.

**Proof of Claim 1.** By Claim 2, the set $\{\tilde{Q}_i^\theta + c_i \mathbf{1} : c_i \in \mathbb{R}\}$ characterizes the solution to equation (25a). Therefore, (25) must have a unique solution $\bar{Q}_i^\theta = \tilde{Q}_i^\theta - \tilde{Q}_i^\theta(\tilde{z}_{N_i^\kappa})\mathbf{1}$. Since (25a) is a overdetermined equation, we can essentially remove one row corresponding to $\tilde{z}_{N_i^\kappa}$, and then plug in $\bar{Q}_i^\theta(\tilde{z}_{N_i^\kappa}) = 0$. This corresponds exactly to the equation in (24). As such, the solution of (24) is the solution of (25), removing the entry in $\tilde{z}_{N_i^\kappa}$. $\square$

By Claim 1 and Claim 2, we have

$$\Phi_i \hat{Q}_i^\theta = \tilde{\Phi}_i \bar{Q}_i^\theta = \tilde{\Phi}_i \tilde{Q}_i^\theta - \tilde{Q}_i^\theta(\tilde{z}_{N_i^\kappa})\mathbf{1}.$$

As such, we can set $c_i^\theta = \tilde{Q}_i^\theta(\tilde{z}_{N_i^\kappa})$ and get,

$$\|\Phi_i \hat{Q}_i^\theta + c_i^\theta \mathbf{1} - Q_i^\theta\|_{D^\theta} = \|\tilde{\Phi}_i \tilde{Q}_i^\theta - Q_i^\theta\|_{D^\theta}. \tag{26}$$

Finally, we bound $\|\tilde{\Phi}_i \tilde{Q}_i^\theta - Q_i^\theta\|_{D^\theta}$. We have, using $\tilde{Q}_i^\theta$ is a fixed point of $g(\cdot)$ and $Q_i^\theta$ is a fixed point of $\mathrm{TD}_i^\theta$,

$$\tilde{\Phi}_i \tilde{Q}_i^\theta - Q_i^\theta = \tilde{\Phi}_i \tilde{Q}_i^\theta - \tilde{\Phi}_i \Pi_i^\theta Q_i^\theta + \tilde{\Phi}_i \Pi_i^\theta Q_i^\theta - Q_i^\theta$$
$$= \tilde{\Phi}_i \Pi_i^\theta \mathrm{TD}_i^\theta \tilde{\Phi}_i \tilde{Q}_i^\theta - \tilde{\Phi}_i \Pi_i^\theta \mathrm{TD}_i^\theta Q_i^\theta + \tilde{\Phi}_i \Pi_i^\theta Q_i^\theta - Q_i^\theta$$
$$= \tilde{\Phi}_i \Pi_i^\theta P^\theta(\tilde{\Phi}_i \tilde{Q}_i^\theta - Q_i^\theta) + \tilde{\Phi}_i \Pi_i^\theta Q_i^\theta - Q_i^\theta.$$

Note that by Assumption 4,

$$\|\tilde{\Phi}_i \Pi_i^\theta P^\theta(\tilde{\Phi}_i \tilde{Q}_i^\theta - Q_i^\theta)\|_{D^\theta} = \|\tilde{\Phi}_i \Pi_i^\theta (P^\theta - \mathbf{1}(\pi^\theta)^\top)(\tilde{\Phi}_i \tilde{Q}_i^\theta - Q_i^\theta)\|_{D^\theta} \le \mu_D \|\tilde{\Phi}_i \tilde{Q}_i^\theta - Q_i^\theta\|_{D^\theta}.$$

This shows $\|\tilde{\Phi}_i \tilde{Q}_i^\theta - Q_i^\theta\|_{D^\theta} \le \mu_D \|\tilde{\Phi}_i \tilde{Q}_i^\theta - Q_i^\theta\|_{D^\theta} + \|\tilde{\Phi}_i \Pi_i^\theta Q_i^\theta - Q_i^\theta\|_{D^\theta}$, and hence,

$$\|\tilde{\Phi}_i \tilde{Q}_i^\theta - Q_i^\theta\|_{D^\theta} \le \frac{1}{1 - \mu_D} \|\tilde{\Phi}_i \Pi_i^\theta Q_i^\theta - Q_i^\theta\|_{D^\theta} \le \frac{1}{1 - \mu_D} \|\tilde{\Phi}_i \Pi_i^\theta Q_i^\theta - Q_i^\theta\|_\infty. \tag{27}$$

Next, recall that the $z_{N_i^\kappa}$'s row of $\Pi_i^\theta$ is the distribution of the state-action pair $z$ conditioned on its $N_i^\kappa$ coordinates being fixed to be $z_{N_i^\kappa}$. We denote this conditional distribution of the states outside of $N_i^\kappa$, $z_{N_{-i}^\kappa}$, given $z_{N_i^\kappa}$, as $\pi^\theta(z_{N_{-i}^\kappa}|z_{N_i^\kappa})$. With this notation,

$$(\tilde{\Phi}_i \Pi_i^\theta Q_i^\theta)(z_{N_i^\kappa}, z_{N_{-i}^\kappa}) = \sum_{z'_{N_{-i}^\kappa}} \pi^\theta(z'_{N_{-i}^\kappa}|z_{N_i^\kappa}) Q_i^\theta(z_{N_i^\kappa}, z'_{N_{-i}^\kappa}).$$

Therefore, we have,

$$|(\tilde{\Phi}_i \Pi_i^\theta Q_i^\theta)(z_{N_i^\kappa}, z_{N_{-i}^\kappa}) - Q_i^\theta(z_{N_i^\kappa}, z_{N_{-i}^\kappa})|$$

$$= \left| \sum_{z'_{N_{-i}^\kappa}} \pi^\theta(z'_{N_{-i}^\kappa}|z_{N_i^\kappa}) Q_i^\theta(z_{N_i^\kappa}, z'_{N_{-i}^\kappa}) - \sum_{z'_{N_{-i}^\kappa}} \pi^\theta(z'_{N_{-i}^\kappa}|z_{N_i^\kappa}) Q_i^\theta(z_{N_i^\kappa}, z_{N_{-i}^\kappa}) \right|$$

$$\leq \sum_{z'_{N_{-i}^\kappa}} \pi^\theta(z'_{N_{-i}^\kappa}|z_{N_i^\kappa}) \left| Q_i^\theta(z_{N_i^\kappa}, z'_{N_{-i}^\kappa}) - Q_i^\theta(z_{N_i^\kappa}, z_{N_{-i}^\kappa}) \right|$$

$$\leq c\rho^{\kappa+1},$$

where the last inequality is due to the exponential decay property (cf. Definition 1 and Assumption 2). Therefore,

$$\|\tilde{\Phi}_i \Pi_i^\theta Q_i^\theta - Q_i^\theta\|_\infty \leq c\rho^{\kappa+1}.$$

Combining the above with (27), we get,

$$\|\tilde{\Phi}_i \tilde{Q}_i^\theta - Q_i^\theta\|_{D^\theta} \leq \frac{c\rho^{\kappa+1}}{1 - \mu_D},$$

which, when combined with (26), leads to the desired result.

## D   Analysis of the Actor and Proof of Theorem 2

The proof is divided into three steps. Firstly, we decompose the error in the gradient approximation into three sequences. Then, we bound the three error sequences seperately. Finally, using the bounds, we prove Theorem 2.

**Step 1: Error decomposition.** Recall that the actor update can be written as $\theta_i(t+1) = \theta_i(t) + \eta_t \Gamma(\hat{\mathbf{Q}}^t) \hat{g}_i(t)$, where

$$\hat{g}_i(t) = \nabla_{\theta_i} \log \zeta_i^{\theta_i(t)}(a_i(t)|s_i(t)) \frac{1}{n} \sum_{j \in N_i^\kappa} \hat{Q}_j^t(s_{N_j^\kappa}(t), a_{N_j^\kappa}(t)),$$

and $\Gamma(\hat{\mathbf{Q}}^t) = \frac{1}{1 + \max_j \|\hat{Q}_j^t\|_\infty}$ is a scalar whose purpose is to control the size of the approximated gradient. We also denote $\Gamma_t = \Gamma(\hat{\mathbf{Q}}^t)$. Recall that the true gradient of the objective function is given by (Lemma 1),

$$\nabla_{\theta_i} J(\theta(t)) = \mathbb{E}_{(s,a)\sim\pi^{\theta(t)}} \nabla_{\theta_i} \log \zeta_i^{\theta_i(t)}(a_i|s_i) \frac{1}{n} \sum_{j \in \mathcal{N}} Q_j^{\theta(t)}(s, a).$$

The error between the approximated gradient $\hat{g}_i(t)$ and the true gradient $\nabla_{\theta_i} J(\theta(t))$ can be decomposed into three terms,

$$\hat{g}_i(t) - \nabla_{\theta_i} J(\theta(t))$$

$$= \nabla_{\theta_i} \log \zeta_i^{\theta_i(t)}(a_i(t)|s_i(t)) \frac{1}{n} \sum_{j \in N_i^\kappa} \hat{Q}_j^t(s_{N_j^\kappa}(t), a_{N_j^\kappa}(t)) - \mathbb{E}_{(s,a)\sim\pi^{\theta(t)}} \nabla_{\theta_i} \log \zeta_i^{\theta_i(t)}(a_i|s_i) \frac{1}{n} \sum_{j \in N_i^\kappa} \hat{Q}_j^t(s_{N_j^\kappa}, a_{N_j^\kappa})$$

$$+ \mathbb{E}_{(s,a)\sim\pi^{\theta(t)}} \nabla_{\theta_i} \log \zeta_i^{\theta_i(t)}(a_i|s_i) \frac{1}{n} \left[ \sum_{j \in N_i^\kappa} \hat{Q}_j^t(s_{N_j^\kappa}, a_{N_j^\kappa}) - \sum_{j \in N_i^\kappa} \hat{Q}_j^{\theta(t)}(s_{N_j^\kappa}, a_{N_j^\kappa}) \right]$$

$$+ \mathbb{E}_{(s,a) \sim \pi^{\theta(t)}} \nabla_{\theta_i} \log \zeta_i^{\theta_i(t)}(a_i|s_i) \frac{1}{n} \Big[ \sum_{j \in N_i^{\kappa}} \hat{Q}_j^{\theta(t)}(s_{N_j^{\kappa}}, a_{N_j^{\kappa}}) - \sum_{j \in \mathcal{N}} Q_j^{\theta(t)}(s, a) \Big]$$

$$:= e_i^1(t) + e_i^2(t) + e_i^3(t).$$

We also use $e^1(t), e^2(t), e^3(t), \hat{g}(t)$ to denote $e_i^1(t), e_i^2(t), e_i^3(t), \hat{g}_i(t)$ stacked into a larger vector respectively. We next bound the three error sequences $e_i^1(t), e_i^2(t)$ and $e_i^3(t)$.

**Step 2: Bounding error sequences.** In this step, we provide bounds on the error sequences. We will frequently use the following auxiliary result, whose proof is omitted as it is identical to that of Lemma 7.

**Lemma 11.** *We have for any $\theta$ and $i$, $\|Q_i^\theta\|_\infty \leq Q_{\max} = \frac{2c_\infty \bar{r}}{1-\mu_D}$. As a result, $\|\nabla_{\theta_i} J(\theta)\| \leq L_i Q_{\max}$ and $\|\nabla J(\theta)\| \leq L Q_{\max}$.*

We start with a bound related to error sequence $e_i^1(t)$, the proof of which is postponed to Appendix D.1.

**Lemma 12.** *Almost surely, for all $i$, we have $\sum_{t=0}^T \eta_t \langle \nabla_{\theta_i} J(\theta(t)), \Gamma_t e_i^1(t) \rangle$ converges to a finite limit as $T \to \infty$.*

Then, we bound error sequence $e_i^2(t)$ in the following Lemma 13, which is an immediate consequence from our analysis of critic in Theorem 3 of Appendix C.

**Lemma 13.** *Almost surely, $\lim_{t \to \infty} e_i^2(t) = 0$.*

*Proof.* We have $\|e_i^2(t)\| \leq L_i \frac{1}{n} \sum_{j \in N_i^\kappa} \|\hat{Q}_j^t - \hat{Q}_j^{\theta(t)}\|_\infty \to 0$ as $t \to \infty$, where we have used part (b) of Theorem 3. $\qquad\square$

Lastly, in Lemma 14 we show that $e_i^3(t)$ can be bounded by a small constant as a result of Theorem 3(a). The proof of Lemma 14 is postponed to Appendix D.2.

**Lemma 14.** *Almost surely, for each $i$, $\|e_i^3(t)\| \leq L_i \frac{c\rho^{\kappa+1}}{1-\mu_D}$.*

With these preparations, we are now ready to prove Theorem 2.

**Step 3: Proof of Theorem 2.** Recall that $\beta_t = \eta_t \Gamma_t$. Note that by the definition of $\Gamma(\cdot)$, $\beta_t \leq \eta_t$. Further, almost surely there exists some constant $\gamma$ s.t. $\beta_t \geq \gamma \eta_t$, as by Theorem 3(b), almost surely, $\|\hat{Q}_j^t\|_\infty$ is uniformly upper bounded for all $t \geq 0, j \in \mathcal{N}$ by some constant. Since the objective function is $L'$-smooth, we have,

$$J(\theta(t+1))$$
$$\geq J(\theta(t)) + \langle \nabla J(\theta(t)), \beta_t \hat{g}(t) \rangle - \frac{L'}{2} \beta_t^2 \|\hat{g}(t)\|^2$$
$$= J(\theta(t)) + \beta_t \|\nabla J(\theta(t))\|^2 + \beta_t \langle \nabla J(\theta(t)), e^1(t) + e^2(t) + e^3(t) \rangle - \frac{L'}{2} \beta_t^2 \|\hat{g}(t)\|^2.$$

Therefore, by a telescope sum we have,

$$J(\theta(T+1)) \geq J(\theta(0)) + \sum_{t=0}^T \beta_t \|\nabla J(\theta(t))\|^2 + \sum_{t=0}^T \eta_t \langle \nabla J(\theta(t)), \Gamma_t e^1(t) \rangle + \sum_{t=0}^T \beta_t \langle \nabla J(\theta(t)), e^2(t) \rangle$$

$$+ \sum_{t=0}^T \beta_t \langle \nabla J(\theta(t)), e^3(t) \rangle - \sum_{t=0}^T \frac{L'}{2} \beta_t^2 \|\hat{g}(t)\|^2$$

$$\geq \sum_{t=0}^T \beta_t \|\nabla J(\theta(t))\|^2 + \sum_{t=0}^T \eta_t \langle \nabla J(\theta(t)), \Gamma_t e^1(t) \rangle - \sum_{t=0}^T \beta_t L Q_{\max} \|e^2(t)\|$$

$$- \sum_{t=0}^T \beta_t \|\nabla J(\theta(t))\| \|e^3(t)\| - \sum_{t=0}^T \frac{L'}{2} \eta_t^2 L^2,$$

where in the last step, we have used $\|\nabla J(\theta(t))\| \leq L Q_{\max}$ (cf. Lemma 11); we have also used that $\|\Gamma_t \hat{g}(t)\| \leq L$. Then, rearranging the above inequality, we get,

$$
\frac{\sum_{t=0}^T \beta_t (\|\nabla J(\theta(t))\|^2 - \|\nabla J(\theta(t))\| \|e^3(t)\|)}{\sum_{t=0}^T \beta_t}
$$

$$
\leq \frac{J(\theta(T+1)) - \sum_{t=0}^T \eta_t \langle \nabla J(\theta(t)), \Gamma_t e^1(t) \rangle + \sum_{t=0}^T \frac{L'}{2} \eta_t^2 L^2}{\gamma \sum_{t=0}^T \eta_t} + L Q_{\max} \frac{\sum_{t=0}^T \beta_t \|e^2(t)\|}{\sum_{t=0}^T \beta_t}
$$

$$
\leq \frac{\bar{r} - \sum_{t=0}^T \eta_t \langle \nabla J(\theta(t)), \Gamma_t e^1(t) \rangle + \sum_{t=0}^T \frac{L'}{2} \eta_t^2 L^2}{\gamma \sum_{t=0}^T \eta_t} + L Q_{\max} \frac{\sum_{t=0}^T \beta_t \|e^2(t)\|}{\sum_{t=0}^T \beta_t}, \tag{28}
$$

where we have used $J(\theta(T+1)) \leq \bar{r}$ (Assumption 1) and $\beta_t \geq \gamma \eta_t$. In (28), when $T \to \infty$, the first term on the right hand side goes to zero as its denominator goes to infinity (Assumption 3) while its nominator is bounded (using Lemma 12 and $\sum_{t=0}^{\infty} \eta_t^2 < \infty$); the second term goes to zero as $\|e^2(t)\| \to 0$ (Lemma 13) and $\sum_{t=0}^T \beta_t \geq \gamma \sum_{t=0}^T \eta_t \to \infty$. So the right hand side of (28) converges to 0. From this, we have by Lemma 14,

$$
\liminf_{t \to \infty} \|J(\theta(t))\| \leq \sup_{t \geq 0} \|e^3(t)\| \leq L \frac{c \rho^{\kappa+1}}{1 - \mu_D},
$$

because otherwise, the left hand side of (28) will be positive and bounded away from zero as $T \to \infty$, a contradction.

### D.1  Proof of Lemma 12

We fix $i$ and define for $z = (s, a)$, $\hat{\mathbf{Q}} = \{\hat{Q}_i\}_{i=1}^n$,

$$
F^{\theta}(\hat{\mathbf{Q}}, z) = \langle \nabla_{\theta_i} J(\theta), \Gamma(\hat{\mathbf{Q}}) \nabla_{\theta_i} \log \zeta_i^{\theta_i}(a_i | s_i) \frac{1}{n} \sum_{j \in N_i^{\kappa}} \hat{Q}_j(s_{N_j^{\kappa}}, a_{N_j^{\kappa}}) \rangle,
$$

and $\bar{F}^{\theta}(\hat{\mathbf{Q}}) = \mathbb{E}_{z \sim \pi^{\theta}} F^{\theta}(\hat{\mathbf{Q}}, z)$. We also define $\hat{F}^{\theta}(\hat{\mathbf{Q}}, \cdot)$ to be the solution of the Poission equation:

$$
\hat{F}^{\theta}(\hat{\mathbf{Q}}, z) = F^{\theta}(\hat{\mathbf{Q}}, z) - \bar{F}^{\theta}(\hat{\mathbf{Q}}) + P^{\theta} \hat{F}^{\theta}(\hat{\mathbf{Q}}, z) = \mathbb{E}_{\theta} \Big[ \sum_{t=0}^{\infty} (F^{\theta}(\hat{\mathbf{Q}}, z(t)) - \bar{F}^{\theta}(\hat{\mathbf{Q}})) \Big| z(0) = z \Big],
$$

where $P^{\theta}$ is the transition kernal on the state-action pair under policy $\theta$, and $P^{\theta} \hat{F}^{\theta}(\hat{\mathbf{Q}}, z) = \mathbb{E}_{z' \sim P^{\theta}(\cdot|z)} \hat{F}^{\theta}(\hat{\mathbf{Q}}, z')$.

One can easily check that $\hat{F}^{\theta}(\cdot, \cdot)$ satisfies the following properties, the proof of which is deferred to the end of this subsection.

**Lemma 15.** *There exists $C_F, L_{\theta,F}, L_{Q,F} > 0$ s.t. for all $\theta, z$, $|\hat{F}^{\theta}(\hat{\mathbf{Q}}, z)| \leq C_F$ and $\hat{F}^{\theta}(\hat{\mathbf{Q}}, z)$ is $L_{\theta,F}$-Lipschitz continuous in $\theta$ in Euclidean norm, and $L_{Q,F}$-Lipschitz continuous in $\hat{\mathbf{Q}}$ in the sense that,*

$$
|\hat{F}^{\theta}(\hat{\mathbf{Q}}, z) - \hat{F}^{\theta}(\hat{\mathbf{Q}}', z)| \leq L_{Q,F} \sum_{j=1}^n \|\hat{Q}_j - \hat{Q}_j'\|_{\infty}.
$$

With this definition, we can decompose $\langle \nabla_{\theta_i} J(\theta(t)), \Gamma_t e_i^1(t) \rangle$ into the following terms,

$$
\begin{aligned}
\langle \nabla_{\theta_i} J(\theta(t)), \Gamma_t e_i^1(t) \rangle &= F^{\theta(t)}(\hat{\mathbf{Q}}^t, z(t)) - \bar{F}^{\theta(t)}(\hat{\mathbf{Q}}^t) \\
&= \hat{F}^{\theta(t)}(\hat{\mathbf{Q}}^t, z(t)) - P^{\theta(t)} \hat{F}^{\theta(t)}(\hat{\mathbf{Q}}^t, z(t)) \\
&= \hat{F}^{\theta(t)}(\hat{\mathbf{Q}}^t, z(t+1)) - P^{\theta(t)} \hat{F}^{\theta(t)}(\hat{\mathbf{Q}}^t, z(t)) \\
&\quad + \hat{F}^{\theta(t-1)}(\hat{\mathbf{Q}}^{t-1}, z(t)) - \hat{F}^{\theta(t)}(\hat{\mathbf{Q}}^t, z(t+1)) \\
&\quad + \hat{F}^{\theta(t)}(\hat{\mathbf{Q}}^t, z(t)) - \hat{F}^{\theta(t-1)}(\hat{\mathbf{Q}}^{t-1}, z(t)) \\
&= a^1(t) + a^2(t) + a^3(t) + a^4(t),
\end{aligned}
$$

where we have defined,

$$a^1(t) = \hat{F}^{\theta(t)}(\hat{\mathbf{Q}}^t, z(t+1)) - P^{\theta(t)}\hat{F}^{\theta(t)}(\hat{\mathbf{Q}}^t, z(t)),$$

$$a^2(t) = \frac{1}{\eta_t}(\eta_{t-1}\hat{F}^{\theta(t-1)}(\hat{\mathbf{Q}}^{t-1}, z(t)) - \eta_t\hat{F}^{\theta(t)}(\hat{\mathbf{Q}}^t, z(t+1))),$$

$$a^3(t) = \frac{\eta_t - \eta_{t-1}}{\eta_t}\hat{F}^{\theta(t-1)}(\hat{\mathbf{Q}}^{t-1}, z(t)),$$

$$a^4(t) = \hat{F}^{\theta(t)}(\hat{\mathbf{Q}}^t, z(t)) - \hat{F}^{\theta(t-1)}(\hat{\mathbf{Q}}^{t-1}, z(t)).$$

With the decomposition, in what follows we show that $\sum_{t=1}^T \eta_t a^j(t)$ converges to a finite limit almost surely for $j = 1, 2, 3, 4$, which together will conclude the proof of this lemma.

For $a^1(t)$, let $\mathcal{F}_t$ be the $\sigma$-algebra generated by $\{\theta(k), \hat{\mathbf{Q}}^k, z(k)\}_{k \le t}$. Then, $a^1(t)$ is $\mathcal{F}_{t+1}$-measurable and $\mathbb{E}a^1(t)|\mathcal{F}_t = 0$. As such, $\sum_{t=1}^T \eta_t a^1(t)$ is a martingale process, and further,

$$\mathbb{E}|\sum_{t=1}^T \eta_t a^1(t)|^2 = \sum_{t=1}^T \eta_t^2 \mathbb{E}|a^1(t)|^2 \le 4C_F^2 \sum_{t=0}^\infty \eta_t^2 < \infty.$$

As such, by martingale convergence theorem, $\sum_{t=1}^T \eta_t a^1(t)$ converges to a finite limit as $T \to \infty$ almost surely.

For $a^2(t)$, note that

$$\sum_{t=1}^T \eta_t a^2(t) = \eta_0 \hat{F}^{\theta(0)}(\hat{\mathbf{Q}}^0, z(1)) - \eta_T \hat{F}^{\theta(T)}(\hat{\mathbf{Q}}^T, z(T+1)),$$

which also converges to a finite limit as $T \to \infty$, almost surely.

For $a^3(t)$, since the step size $\eta_t$ is non-increasing, we have,

$$\sum_{t=1}^T \eta_t|a^3(t)| = \sum_{t=1}^T (\eta_{t-1} - \eta_t)|\hat{F}^{\theta(t-1)}(\hat{\mathbf{Q}}^{t-1}, z(t))| \le C_F(\eta_0 - \eta_T) < C_F\eta_0.$$

As such $\sum_{t=1}^T \eta_t a^3(t)$ converges to a finite limit almost surely.

Finally, for $a^4(t)$, we note that by the Lipschitz property of $\hat{F}^\theta(\hat{\mathbf{Q}}, z)$ in Lemma 15,

$$|a^4(t)| \le |\hat{F}^{\theta(t)}(\hat{\mathbf{Q}}^t, z(t)) - \hat{F}^{\theta(t-1)}(\hat{\mathbf{Q}}^t, z(t))| + |\hat{F}^{\theta(t-1)}(\hat{\mathbf{Q}}^t, z(t)) - \hat{F}^{\theta(t-1)}(\hat{\mathbf{Q}}^{t-1}, z(t))|$$

$$\le L_{\theta,F}\|\theta(t) - \theta(t-1)\| + L_{Q,F}\sum_{j=1}^n \|\hat{Q}_j^t - \hat{Q}_j^{t-1}\|_\infty$$

$$\le \bar{L}_{\theta,F}\eta_{t-1} + \bar{L}_{Q,F}\alpha_{t-1},$$

for some constant $\bar{L}_{\theta,F}$ and $\bar{L}_{Q,F}$ almost surely. Here we have used $\|\theta(t) - \theta(t-1)\| \le \eta_{t-1}L$ (check how we verified Assumption 9 in Appendix C.2). Further, we have used $\|\hat{Q}_j^t - \hat{Q}_j^{t-1}\|_\infty \le \alpha_{t-1}(2\bar{r} + 2\|\hat{Q}_j^{t-1}\|_\infty)$ (cf. equation (16) in Appendix C.2), and the fact that $\|\hat{Q}_j^{t-1}\|_\infty$ is upper bounded uniformly over $t$ almost surely, cf. Theorem 3. As such, we have

$$\sum_{t=1}^T \eta_t|a^4(t)| \le \sum_{t=1}^\infty \left(\bar{L}_{\theta,F}\eta_t\eta_{t-1} + \bar{L}_{Q,F}\alpha_{t-1}\eta_t\right) < \infty.$$

As a result, we have, $\sum_{t=1}^T \eta_t a^4(t)$ converges to a finite limit as $T \to \infty$, almost surely. This concludes the proof of Lemma 12.

Finally, we provide the proof for Lemma 15.

*Proof of Lemma 15.* Clearly, $|F^\theta(\hat{\mathbf{Q}}, z)| \le \|\nabla_{\theta_i} J(\theta)\| L_i \le L_i^2 Q_{\max} := C_F'$, where we have used $\|\nabla_{\theta_i} J(\theta)\| \le L_i Q_{\max}$ (cf. Lemma 11). Using the same argument as in Lemma 7 (b), we have $|\hat{F}^\theta(\hat{\mathbf{Q}}, z)| \le C_F = \frac{2c_\infty}{1-\mu_D}C_F'$. Next, note that,

$$|F^\theta(\hat{\mathbf{Q}}, z) - F^{\bar{\theta}}(\hat{\mathbf{Q}}, z)|$$

$$\leq |\langle \nabla_{\theta_i} J(\theta) - \nabla_{\theta_i} J(\bar{\theta}), \Gamma(\hat{\mathbf{Q}}) \nabla_{\theta_i} \log \zeta_i^{\theta_i}(a_i|s_i) \frac{1}{n} \sum_{j \in N_i^\kappa} \hat{Q}_j(s_{N_j^\kappa}, a_{N_j^\kappa}) \rangle|$$

$$+ |\langle \nabla_{\theta_i} J(\bar{\theta}), \Gamma(\hat{\mathbf{Q}}) (\nabla_{\theta_i} \log \zeta_i^{\theta_i}(a_i|s_i) - \nabla_{\theta_i} \log \zeta_i^{\bar{\theta}_i}(a_i|s_i)) \frac{1}{n} \sum_{j \in N_i^\kappa} \hat{Q}_j(s_{N_j^\kappa}, a_{N_j^\kappa}) \rangle|$$

$$\leq L' L_i \|\theta - \bar{\theta}\| + L_i Q_{\max} L_i' \|\theta - \bar{\theta}\| := L_{\theta,F}' \|\theta - \bar{\theta}\|.$$

The above shows $F^\theta(\hat{\mathbf{Q}}, z)$ is Lipschitz in $\theta$. Then, using a similar argument as Lemma 8 (c), we can show $\hat{F}^\theta(\hat{\mathbf{Q}}, z)$ is Lipschitz continuous in $\theta$. To do this, we fix any initial $z$, let $d^{\theta,t}$ be the distribution of $z(t)$ under policy $\theta$. Then, equation (22) in the proof of Lemma 8 shows that $d^{\theta,t} - \pi^\theta$ is Lipschitz in $\theta$ with Lipschitz constant geometrically decaying in $t$, i.e. for some $L_d > 0$,

$$\|(d^{\theta,t} - \pi^\theta) - (d^{\bar{\theta},t} - \pi^{\bar{\theta}})\|_1 \leq L_d (\frac{\mu_D + 1}{2})^t \|\theta - \bar{\theta}\|.$$

Note that by definition,

$$\hat{F}^\theta(\hat{\mathbf{Q}}, z) = \mathbb{E}_\theta \Big[ \sum_{t=0}^\infty (F^\theta(\hat{\mathbf{Q}}, z(t)) - \bar{F}^\theta(\hat{\mathbf{Q}})) \Big| z(0) = z \Big]$$

$$= \sum_{t=0}^\infty [\mathbb{E}_{z' \sim d^{\theta,t}} F^\theta(\hat{\mathbf{Q}}, z') - \mathbb{E}_{z' \sim \pi^\theta} F^\theta(\hat{\mathbf{Q}}, z')]$$

$$= \sum_{t=0}^\infty \sum_{z' \in \mathcal{Z}} (d^{\theta,t}(z') - \pi^\theta(z')) F^\theta(\hat{\mathbf{Q}}, z'). \tag{29}$$

As such,

$$|\hat{F}^\theta(\hat{\mathbf{Q}}, z) - \hat{F}^{\bar{\theta}}(\hat{\mathbf{Q}}, z)| \leq \sum_{t=0}^\infty \sum_{z' \in \mathcal{Z}} \Big| (d^{\theta,t}(z') - \pi^\theta(z')) F^\theta(\hat{\mathbf{Q}}, z') - (d^{\bar{\theta},t}(z') - \pi^{\bar{\theta}}(z')) F^{\bar{\theta}}(\hat{\mathbf{Q}}, z') \Big|$$

$$\leq \sum_{t=0}^\infty \sum_{z' \in \mathcal{Z}} \Big[ |d^{\theta,t}(z') - \pi^\theta(z') - (d^{\bar{\theta},t}(z') - \pi^{\bar{\theta}}(z'))| |F^\theta(\hat{\mathbf{Q}}, z')|$$

$$+ |d^{\bar{\theta},t}(z') - \pi^{\bar{\theta}}(z')| |F^\theta(\hat{\mathbf{Q}}, z') - F^{\bar{\theta}}(\hat{\mathbf{Q}}, z')| \Big]$$

$$\leq \sum_{t=0}^\infty \Big[ \|d^{\theta,t} - \pi^\theta - (d^{\bar{\theta},t} - \pi^{\bar{\theta}})\|_1 C_F' + \|d^{\bar{\theta},t} - \pi^{\bar{\theta}}\|_1 L_{\theta,F}' \|\theta - \bar{\theta}\| \Big]$$

$$\leq \sum_{t=0}^\infty \Big[ L_d C_F' (\frac{1 + \mu_D}{2})^t \|\theta - \bar{\theta}\| + 2 c_\infty \mu_D^t L_{\theta,F}' \|\theta - \bar{\theta}\| \Big]$$

$$\leq L_{\theta,F} \|\theta - \bar{\theta}\|,$$

for some $L_{\theta,F} > 0$. This shows that $\hat{F}^\theta(\hat{\mathbf{Q}}, z)$ is Lipschitz continuous in $\theta$.

Finally, we show $\hat{F}^\theta(\hat{\mathbf{Q}}, z)$ is Lipschitz in $\hat{\mathbf{Q}}$. Note that,

$$|F^\theta(\hat{\mathbf{Q}}, z) - F^\theta(\hat{\mathbf{Q}}', z)| \leq L_i^2 Q_{\max} \Big| \Gamma(\hat{\mathbf{Q}}) \frac{1}{n} \sum_{j \in N_i^\kappa} \hat{Q}_j(s_{N_j^\kappa}, a_{N_j^\kappa}) - \Gamma(\hat{\mathbf{Q}}') \frac{1}{n} \sum_{j \in N_i^\kappa} \hat{Q}_j'(s_{N_j^\kappa}, a_{N_j^\kappa}) \Big|$$

$$\leq L_i^2 Q_{\max} \Big| (\Gamma(\hat{\mathbf{Q}}) - \Gamma(\hat{\mathbf{Q}}')) \frac{1}{n} \sum_{j \in N_i^\kappa} \hat{Q}_j(s_{N_j^\kappa}, a_{N_j^\kappa}) \Big|$$

$$+ L_i^2 Q_{\max} \Gamma(\hat{\mathbf{Q}}') \Big| \frac{1}{n} \sum_{j \in N_i^\kappa} (\hat{Q}_j(s_{N_j^\kappa}, a_{N_j^\kappa}) - \hat{Q}_j'(s_{N_j^\kappa}, a_{N_j^\kappa})) \Big|.$$

Note that

$$|\Gamma(\hat{\mathbf{Q}}) - \Gamma(\hat{\mathbf{Q}}')| = \frac{|\max_j \|\hat{Q}_j\|_\infty - \max_j \|\hat{Q}_j'\|_\infty|}{(1 + \max_j \|\hat{Q}_j\|_\infty)(1 + \max_j \|\hat{Q}_j'\|_\infty)} \leq \frac{\sum_j \|\hat{Q}_j - \hat{Q}_j'\|_\infty}{(1 + \max_j \|\hat{Q}_j\|_\infty)}.$$

As such,

$$|F^\theta(\hat{\mathbf{Q}}, z) - F^\theta(\hat{\mathbf{Q}}', z)| \le L_i^2 Q_{\max} \sum_j \|\hat{Q}_j - \hat{Q}_j'\|_\infty + L_i^2 Q_{\max} \frac{1}{n} \sum_j \|\hat{Q}_j - \hat{Q}_j'\|_\infty$$

$$:= L'_{Q,F} \sum_j \|\hat{Q}_j - \hat{Q}_j'\|_\infty,$$

which shows $F^\theta(\hat{\mathbf{Q}}, z)$ is Lipschitz in $\hat{\mathbf{Q}}$. Then, by (29),

$$|\hat{F}^\theta(\hat{\mathbf{Q}}, z) - \hat{F}^\theta(\hat{\mathbf{Q}}', z)| \le \sum_{t=0}^\infty \sum_{z' \in \mathcal{Z}} |d^{\theta,t}(z') - \pi^\theta(z')| \Big| F^\theta(\hat{\mathbf{Q}}, z') - F^\theta(\hat{\mathbf{Q}}', z') \Big|$$

$$\le \sum_{t=0}^\infty \|d^{\theta,t} - \pi^\theta\|_1 L'_{Q,F} \sum_j \|\hat{Q}_j - \hat{Q}_j'\|_\infty$$

$$\le \sum_{t=0}^\infty 2c_\infty \mu_D^t L'_{Q,F} \sum_j \|\hat{Q}_j - \hat{Q}_j'\|_\infty$$

$$\le L_{Q,F} \sum_j \|\hat{Q}_j - \hat{Q}_j'\|_\infty,$$

which shows $\hat{F}^\theta(\hat{\mathbf{Q}}, z)$ is Lipschitz in $\hat{\mathbf{Q}}$. $\qquad\qquad\qquad\qquad\qquad\qquad\square$

### D.2   Proof of Lemma 14

Let $c_j^{\theta(t)}$ be the constant in Theorem 3(a). Then,

$$e_i^3(t) = \mathbb{E}_{(s,a)\sim\pi^{\theta(t)}} \nabla_{\theta_i} \log \zeta_i^{\theta_i(t)}(a_i|s_i) \frac{1}{n} \Big[ \sum_{j \in N_i^\kappa} \hat{Q}_j^{\theta(t)}(s_{N_j^\kappa}, a_{N_j^\kappa}) - \sum_{j=1}^n Q_j^{\theta(t)}(s,a) \Big]$$

$$= \mathbb{E}_{(s,a)\sim\pi^{\theta(t)}} \nabla_{\theta_i} \log \zeta_i^{\theta_i(t)}(a_i|s_i) \frac{1}{n} \Big[ \sum_{j=1}^n \hat{Q}_j^{\theta(t)}(s_{N_j^\kappa}, a_{N_j^\kappa}) - \sum_{j=1}^n Q_j^{\theta(t)}(s,a) \Big]$$

$$= \frac{1}{n} \sum_{j=1}^n \mathbb{E}_{(s,a)\sim\pi^{\theta(t)}} \nabla_{\theta_i} \log \zeta_i^{\theta_i(t)}(a_i|s_i) \Big[ \hat{Q}_j^{\theta(t)}(s_{N_j^\kappa}, a_{N_j^\kappa}) + c_j^{\theta(t)} - Q_j^{\theta(t)}(s,a) \Big],$$

where in the second equality, we have used for all $j \notin N_i^\kappa$,

$$\mathbb{E}_{(s,a)\sim\pi^{\theta(t)}} \nabla_{\theta_i} \log \zeta_i^{\theta_i(t)}(a_i|s_i) \hat{Q}_j^{\theta(t)}(s_{N_j^\kappa}, a_{N_j^\kappa}) = 0,$$

and in the third equality, we have used for all $j$, $\mathbb{E}_{(s,a)\sim\pi^{\theta(t)}} \nabla_{\theta_i} \log \zeta_i^{\theta_i(t)}(a_i|s_i) c_j^{\theta(t)} = 0$. The reason of these is due to $\hat{Q}_j^{\theta(t)}(s_{N_j^\kappa}, a_{N_j^\kappa})$ does not depend on $a_i$ when $j \notin N_i^\kappa$, and $c_j^{\theta(t)}$ does not depend on $a_i$ for all $j$. For more details, see (9) in the proof of Lemma 2 in Appendix A.1.

As such, by Cauchy Schwarz inequality and Theorem 3(a),

$$\|e_i^3(t)\| \le \frac{1}{n} \sum_{j=1}^n \sqrt{\mathbb{E}_{(s,a)\sim\pi^{\theta(t)}} \|\nabla_{\theta_i} \log \zeta_i^{\theta_i(t)}(a_i|s_i)\|^2} \sqrt{\mathbb{E}_{(s,a)\sim\pi^{\theta(t)}} \Big[ \hat{Q}_j^{\theta(t)}(s_{N_j^\kappa}, a_{N_j^\kappa}) + c_j^{\theta(t)} - Q_j^{\theta(t)}(s,a) \Big]^2}.$$

$$\le L_i \frac{c\rho^{\kappa+1}}{1 - \mu_D}.$$

## Footnotes

[3]Our stochastic approximation scheme (14) is slightly different from Konda and Tsitsiklis (2003) in that in Konda and Tsitsiklis (2003), $h^{\theta(t)}(\cdot)$ and $G^{\theta(t)}(\cdot)$ depend on $z(t+1)$ instead of $z(t)$. This change is without loss of generality as we can group two states togethoer, i.e. $y(t) = (z(t-1), z(t))$ and write our algorithm in the form of Konda and Tsitsiklis (2003).