[Reviews · NeurIPS 2020]

Review 1

Summary and Contributions: This paper presents the counterpart of the results of Qu, Wierman and Li (2019) for average reward MARL (the work of Qu et al. (2019) discusses discounted reward MARL). The main theoretical result is Theorem 1. The paper also presents a Scalable Actor-Critic Algorithm based on the theoretical results.

Strengths: The theoretical results seem to be sound. The topic should be of interest to some researchers in MARL.

Weaknesses: The main weakness in my opinion is the significance of the work. While the setting of discounted reward MARL sounds natural and readily acceptable to me, I would like to be convinced that average reward MARL can model many meaning situations. Unfortunately arguments along this line are not strong in the paper. Even for the example of APs and users in Section 4, I do not see why it should be an average award case. Further comments after rebuttal: I have read the authors' responses and reviews from other reviewers. I believe the authors could have included the points in their rebuttal in the submission. Actually, I wish to see even stronger justifications than the authors have written in the rebuttal.

Correctness: Yes generally. However, it is disappointing to see that the authors do not explain the conclusions from the experiment, particularly Figure 2, in the main text. I agree that space constraints are always an issue for NIPS paper :-), but I believe that readers should not be required to read the appendices and supplementary materials unless they want to know the tedious details or read the complicated proofs.

Clarity: No really. The abstract is a bit misleading. The significance of the work (consideration of the average reward cases) and the results (theorems 1 and 2) could have been more clearly not stated and argued.

Relation to Prior Work: It would be more interesting to compare and contrast the new results in this paper with the counterparts in the work of Qu et al. (2019).

Reproducibility: Yes

Additional Feedback: I hope this paper can be rewritten so that the main points, in particular its significance, can be emphasised.


Review 2

Summary and Contributions: The paper introduces scalable actor-critic (SAC) method for networked MARL where agent's values are dependent on the local interaction with nearby agents. It aims to maximize the global average expected reward per time step instead of the more popular RL objective of maximizing expected discounted reward. It is shown that by exploting exponential decay property SAC learns near optimal individual policy for optimizing the average reward and its complexity scale with state-action space size of local neighbourhood. POST REBUTTAL: I have read other reviews and authors response and they have addressed my comments. I am convinced with their explanation about the difference between underlying framework of the proposed method and that of mean field RL. Also, there are very few MARL literature which deals with average reward settings. Overall, I am retaining my score.

Strengths: The novelty of the paper is to provide a scalable learning method for average reward settings with guarantee of small performance loss. The work is relevant to a number of real world applications such as social networks, communication networks, transportation networks etc. Following are the highlights of the paper - The problem formulation is clear and despite having so many variables in the proofs, the mathematical notations are wisely chosen and are unambiguous. - Error bound is provided on the truncated Q-function using exponential decay property. - Policy gradient is approximated using gradients of localized policy. - The error between true gradient and approximated gradient is shown to be bounded. - It is also shown that the exponential decay property holds true in case of discounted reward under the assumptions. - Convergence guarantee has been provided for the proposed SAC algorithm - Experiment result on a wireless network with multiple access points show that SAC significantly outperforms a baseline based on localized ALOHA protocol. The proofs appears to be correct and I liked the way few assumptions have been used to provide theoretical guarantees. Overall I think the paper should be accepted for publication.

Weaknesses: - Why J(\theta) has been deducted from immediate reward to compute for the Q-values (line 136)? A citation or explanation will be helpful. - To compute approximate gradient (Algorithm 1, line 7), it is assumed that agents have access to Q-values of neighbours, which is not clearly mentioned in the paper. - ALOHA protocol is used as baseline which seems too simplified to me. Comparision with other learning algorithm suited for average reward setting will make the results more convincing.

Correctness: The technical content of the paper appears correct to me. The authors have used exponential decay property nicely to show that the Q values and gradients can be efficiently approximated.

Clarity: The paper is well written and is easy to follow.

Relation to Prior Work: Authors have discussed about competitive MARL and independent learning in multi-agent setting and have explained that their work is relevant to cooperative MARL with a more structured setting where each agent has its own state that it acts upon. Also the work deals with maximizing expecetd average reward which is different than other existing work dealing with discounted reward. I feel discussion on existing work which deals with local interaction of agents (for example mean field RL) should be included.

Reproducibility: Yes

Additional Feedback:


Review 3

Summary and Contributions: The paper is concerned with interaction graphs that factor the transition model in multi-agent RL. The authors extend the set for which the Exponential Decay Property holds (Th.1) and prove tighter bounds for these MDP (Co.1). Based on these insights, they propose a tabular average reward variant of the Scalable Actor-Critic algorithm (confusingly named SAC) and prove that the it converges in the limit exponentially with the considered graph-distance $\kappa$. The algorithm is evaluated in a simple multiple access point experiment, which shows that SAC outperforms localized ALOHA, and that considering at least one hop ($\kappa=1$) improves over decentralized value functions ($\kappa=0$). The paper appears to be a straight extension of Qu et al. (2019) to the average reward case, with the exception of Theorem and Corollary 1, which seem to be genuinely novel. POST-REBUTTAL: I have read the reviews and author response. The authors have addressed my questions in the most part sufficiently. In my view Theorem 2 does not really depend on Theorem 1: it assumes the exponential decay property, which Theorem 1 only widens. But if the authors are willing to clarify this, I am satisfied. Based on the rebuttal I raised my score to 7.

Strengths: The discussion of the average reward case of interaction graphs is quite interesting and Theorem/Corollary 1 appears a valuable contribution. The paper contains extensive additional analysis in the supplementary material.

Weaknesses: The reviewer has not read Qu et al. (2019) and did not check the detailed proof, but at first glance Theorem 2 appears somewhat derivative, as it apparently does not make use of the insights of Theorem 1. Furthermore, while tabular methods are an excellent way for theoretical analysis, the paper would have been much stronger with a neural network implementation of Algorithm 1.

Correctness: The reviewer only had time to check the proof of Theorem 1, which is mostly correct. The theorem requires the additional assumption $\bar r \geq 0$, though, as the term can not be moved out of the absolute operator otherwise. As the authors attached 22 pages of proofs and further analysis, it might be better to submit to a journal where reviewers generally have more time checking proofs.

Clarity: The paper is well written, but it is not easy to see the immediate relationship between sections 3.1 and 3.2. It should also clarify the differences to Qu et al. (2019) better, who also define an algorithm called SAC. The authors should make explicit which parts of Theorem 2 are from prior publications and introduce a unique name for the presented method. On a side note, SAC also stands for Soft Actor-Critic (https://arxiv.org/abs/1801.01290), one of the most popular algorithms in deep RL, which increases the confusion. Lastly, Figure 5 should be in the main paper (instead of Figure 1), as it adds substantial insight (even though it is the same insight as Figure 1 in Qu et al., 2019).

Relation to Prior Work: The related literature section discusses a wide range of related works, but the reviewer was missing current works on collaborative MARL with neural network approximation. For an overview, the authors are referred to: Oroojlooy Jadid, A. and Hajinezhad, D. A review of cooperative multi-agent deep reinforcement learning. CoRR,abs/1908.03963, 2019. URL http://arxiv.org/ abs/1908.03963 The reviewer was also missing a discussion of the similarities to Sparse Collaborative Q-learning: Kok, J. R. and Vlassis, N. Collaborative multiagent reinforcement learning by payoff propagation. Journal of Machine Learning Research, 7(Sep):1789–1828, 2006.

Reproducibility: Yes

Additional Feedback: The broader impact statement could be more fleshed out: which assumptions would be violated, e.g., in a smart infrastructure system? What could go wrong if SAC is still applied?


Review 4

Summary and Contributions: This paper is concerned with the scalability of multiagent reinforcement learning. It proves that local interactions and exponential decay can be exploited in order to build scalable MARL methods for networked systems with local neighbor approximation. It proposes a scalable actor-critic method with convergence guarantee, and performs numerical experiments in a wireless network environment.

Strengths: This paper proposes novel formulations of MARL in a graph with localized property and exponential decay, and identifies a general condition so that the exponential decay holds. Both of them come with concrete proof. This paper also proposes a scalable actor-critic method based on aforementioned theory, with convergence guarantee.

Weaknesses: While this paper stresses the scalability and richness of the proposed method, the numerical experiment, howeer is very limited and only contains one wireless network scenario with 5x5 grid. the number of users are not large scale and also no baselines are provided in order to compare the performance. As such, it is very difficult to assess the main claim from the paper.

Correctness: Whiel the method is derived from previous theortical work and constains certain novelty theoretically, the empirical experiment is, however, inadequate and does not support the claim due to the following reasons: 1) lack of strong baselines, suggest to compare the mean-field based solution (such as Gu et al 2020), which is certainly efficient in the grid setting. should also include indepent learner and factorised methods. 2) the tasks chosen are limited. check Yang, Yaodong, et al. "Mean field multi-agent reinforcement learning." ICML18 for varous more challenging tasks. 3) I will be keen to see more thorough ablation studies on such as a) varous network structures b) number of agents.

Clarity: mostly clear. a minor comment: 1) Eq. 2 is miss-leading. It is not multiagent formulation but rather a single agent formuation as it assumes there is a global optimizer who sees everything and optimimzes the joint policy \theta.

Relation to Prior Work: Yes. some important relevant work on mean-field approach to multiagent learning is missing. The paper should compare with them both theoretically and experimentally.

Reproducibility: Yes

Additional Feedback: In the numerical experiment, is it possible to simply increase the size of the grid (like 500x500 or even larger) to show the scalibility of the proposed method and compare it against other strong baselines such as mean-field approaches?

[Author Response · NeurIPS 2020]

**Review 1.** Thanks for your comment. In fact, average reward RL is in many applications more relevant than discounted reward and is typically more challenging to study. See the detailed reasons below.

- Application wise, in many real-world networked systems, there is no naturally defined starting state or discounting factor, and the performance is measured under the stationary distribution. For example, in the wireless communication, the long standing performance metric is the throughput, which measures the average number of packets sent under the stationary distribution.
- In RL, it is widely known that studying the average reward is a more challenging topic. For example, the Bellman operator is no longer a contraction in general (because there is no discounting factor), and the set of fixed points is no longer unique, but a subspace instead. From a historic context, it was long known the average reward required very different algorithms (e.g. the $Q$ function becomes the "differential" $Q$-function) and analysis techniques, and there had been a series of work discussing the average reward case, e.g. [Tsitsiklis and Van Roy 1999] cited in the paper, and "John Tsitsiklis and Benjamin Van Roy. On average versus discounted reward temporal-difference learning. Machine Learning 49.2-3 (2002): 179-191".
- On top of the above, when coupled with the multi-agent setting, the average reward case brings additional challenges. Specifically, as shown in Appendix A.2 in the paper, our average reward problem captures certain NP-hard instances. Similar complexity results can be found in [Blondel and Tsitsiklis 2000].

**Review 2.** Thanks for your comment and please find the response to your questions below.

- ($Q$-function) In the definition for the $Q$-function, the cost $J(\theta)$ is subtracted. This is the standard definition for $Q$-function for the average reward case, and is sometimes called differential $Q$-function. The relevant reference can be found in standard textbooks like [Bertsekas 2007] and we will add that to the final version.
- (**Communication**) Yes, access to neighbor's information is needed and we will clarify that in the final paper.
- (**Comparison**) We are happy to add comparison to other learning algorithms. In fact, Appendix E in our supplementary material includes a run of our algorithm with $\kappa = 0$, which is essentially the independent learner approach in the literature. If the reviewer has suggestions on specific algorithms to compare, we are happy to test those.

**Review 3.** Thanks for your comment and please find the response to your questions below.

- (**Thm 2 vs [Qu et al 2019]**) Average reward RL is in general more challenging than the discounted case, see our response to Review 1. Regarding Thm 2 specifically, it actually relies on very different techniques than [Qu et al 2019]. In [Qu et al 2019], at each time, there is a (long) inner loop of critic (TD-learning) steps that estimates the $Q$-function to a good accuracy during which the policy is *fixed*. This makes the analysis of the critic "decoupled" from the policy updates (actor). In the current paper, there is no inner loop, and critic (TD-learning) and actor (policy update) steps are performed *simultaneously*. This creates many challenges in the analysis, as the critic no longer operates under a fixed policy, and the analysis of it cannot be decoupled from the actor.
- (**Relationship between Thm 1 and Thm 2.**) The result of Thm 1 is stated as Assumption 2 for Theorem 2, and as such Thm 2 will surely rely on Thm 1. In terms of proof, the result of Thm 1 is used in Lemma 10, which shows the fixed point of the critic is a good approximation of the full $Q$-function. We make such a separation because (a) Theorem 1 can be useful for many types of RL methods, of which Algo 1/Thm 2 is just one particular actor-critic method; (b) The condition in Thm 1 might be conservative and the exponential decay will hold broadly beyond the condition in Thm 1 (cf. Appendix B.4). So writing the exponential decay as an assumption will broaden the applicability of Thm 2.
- (**Other**) Thanks for the suggestion on neural network implementation and we are happy to try that. We will also incorporate your other suggestions to improve clarity.

**Review 4.** Thanks for your comment and please find our response below.

- (**Comparison with mean-field**) From a theoretic view point, our model and framework is very different from the mean-field approach. While both approaches attempt to address the curse of dimensionality, our framework allows heterogeneous agents, while mean-field approaches typically assume homogeneous agents, whose affect can be well approximated by their mean. Nevertheless, we agree that mean-field approaches could be quite good for grid settings. We are happy to add the above discussions in the revised paper. Also, we are happy to compare our approach with mean-field approach in large scale networks, for both homogeneous agents and heterogeneous agents.
- (**Simulations**) In our supplementary material (Appendix E), we also ran our algorithm with $\kappa = 0$ (each agent runs a single agent actor critic method), which is similar in spirit to the independent learner approach that the reviewer suggested for comparison. That being said, we are happy to incorporate more simulations, including larger number of agents (500*500), different graphs (less "regular" than grid), tasks like Battle games, and comparison with other algorithms (mean field).

[Meta-Review · NeurIPS 2020]

The paper has been extensively discussed and reviewers agree the paper has merit and the rebuttal brings a lot of clarification on a number of questions identified by the reviewers (e.g. the difference between underlying framework of the proposed method and that of mean field RL). General consensus is to propose acceptance of the paper; reviewers would like the authors to clarify the following in the paper though: In difference to their claim, Theorem 2 does not really depend on Theorem 1, as it only assumes the exponential decay property, which Theorem 1 only widens.